# LLmFPCA-detect: LLM-powered Multivariate Functional PCA for Anomaly Detection in Sparse Longitudinal Texts

## Abstract

Sparse longitudinal (SL) textual data arises when individuals generate text repeatedly over time (e.g., customer reviews, occasional social media posts, electronic medical records across visits), but the frequency and timing of observations vary across individuals. These complex textual data sets have immense potential to inform future policy and targeted recommendations. However, because SL text data lack dedicated methods and are noisy, heterogeneous, and prone to anomalies, detecting and inferring key patterns is challenging. We introduce LLmFPCA-detect, a flexible framework that pairs LLM-based text embeddings with functional data analysis to detect clusters and infer anomalies in large SL text datasets. First, LLmFPCA-detect embeds each piece of text into an application-specific numeric space using LLM prompts. Sparse multivariate functional principal component analysis (mFPCA) conducted in the numeric space forms the workhorse to recover primary population characteristics, and produces subject-level scores which, together with baseline static covariates, facilitate data segmentation, unsupervised anomaly detection and inference, and enable other downstream tasks. In particular, we leverage LLMs to perform dynamic keyword profiling guided by the data segments and anomalies discovered by LLmFPCA-detect, and we show that cluster-specific functional PC scores from LLmFPCA-detect, used as features in existing pipelines, help boost prediction performance. We support the stability of LLmFPCA-detect with experiments and evaluate it on two different applications using public datasets, Amazon customer-review trajectories, and Wikipedia talk-page comment streams, demonstrating utility across domains and outperforming state-of-the-art baselines.

## 1 Introduction

In modern machine learning, it is common to encounter datasets comprising of $N$ subjects, where each subject $i$ is associated with a sequence of textual observations $\{K_i(T_{i1}), K_i(T_{i2}), \ldots, K_i(T_{iN_i})\}$ recorded at sparse and irregular time points $\{T_{i1}, T_{i2}, \ldots, T_{iN_i}\} \subset \mathbb{R}$. Despite LLMs having spurred many advancements in analysis of text data, current methods are not well adapted to sparse longitudinal (SL) designs—time-evolving texts observed at irregular, subject-specific times—so these are frequently discarded or collapsed across time, ignoring the dynamic patterns in the texts. In this paper, we propose a novel framework for the analysis of SL text data that yields representations suitable for straightforward integration into unsupervised and supervised learning pipelines. The proposed methodology is applicable to a wide range of domains that generate SL text data, such as, electronic medical records in healthcare (Ford et al., 2016), consumer interactions through service channels in business (Cavique et al., 2022), activity logs from online learning platforms in education (Yang & Kang, 2020), user posts and comments on social media Hutto et al. (2013); Valdez et al. (2020); Kelley & Gillan (2022) and many more.

A major challenge with SL text datasets is that observations are unstructured and noisy, heterogeneous across subjects, and may contain outliers. The first step in making such data amenable for downstream supervised or unsupervised learning tasks, including prediction and inference, is to extract parsimonious feature representations of the longitudinal texts that capture the leading modes of variation. In this work, we propose LLmFPCA-detect, which starts from noisy SL texts and produces learned

representations, accounting for heterogeneity and providing type-I-error–controlled outlier screening. LLmFPCA-detect begins by embedding text into an application specific numeric space using LLMs. In this numeric space, sparse multivariate functional principal component analysis(mFPCA) Happ & Greven (2018); Yao et al. (2005) is used to model the longitudinal text embeddings as noisy observations of an underlying smooth trajectory. The method first clusters the preliminary FPC scores, augmented with baseline subject-level covariates, and then screens for outliers; a novel calibration step yields the final set of anomalies with statistical significance guarantees. We illustrate this new approach on two datasets: the Amazon review corpus and the Wikipedia talk- page comment stream, where LLmFPCA-detect reveals insightful findings from SL text data.

**Related Works**   *Modeling SL data* Beginning with the seminal parametric random-effects formulation Laird & Ware (1982), the field of longitudinal data analysis has undergone extensive development over the decades; see Verbeke et al. (2014) for a review on multivariate longitudinal data analysis. Functional data analysis (FDA) provides a nonparametric framework for SL data—via principal components through conditional expectation Yao et al. (2005); Happ & Greven (2018)—to predict subject-specific smooth trajectories even from one or a few observations. While this line of work has expanded to include dynamic Hao et al. (2024); Zhou & Mueller (2024) and covariate-dependent Kim et al. (2023) extensions, and has led to methods for clustering and unsupervised anomaly detection Schmutz et al. (2020); Wu et al. (2023); Castrillón-Candás & Kon (2022), and supervised tasks such as regression and classification Müller (2005), none of these methods extend directly to heterogeneous, complex SL text data paired with baseline covariates and containing outliers.

*Text time series versus SL texts* An SL design differs from a time series; instead of a single, regularly spaced sequence of observations, it comprises many subjects, each with its own trajectory recorded at irregular, subject-specific times where per-subject sampling is sparse, and between-subject heterogeneity could be substantial. While text time-series modeling has advanced considerably O'Connor et al. (2010); Blei & Lafferty (2006); Wang & McCallum (2006); Bamler & Mandt (2017); Dodds et al. (2011); Griffiths & Steyvers (2004); Yurochkin et al. (2019), these approaches rely on dense, uniformly spaced observations and are not suited to SL texts.

*Anomaly detection* Text clustering and anomaly detection are central NLP tasks, used to flag harmful content, phishing, and spam. With pretrained language models (e.g., BERT Devlin et al. (2019), RoBERTa Liu et al. (2019), GPT Brown et al. (2020)), embedding-based detectors have proliferated alongside other approaches Yin & Wang (2016); Cao et al. (2025); Ruff et al. (2019); Subakti et al. (2022); Dhillon & Modha (2001); Liu et al. (2008); Kannan et al. (2017). Yet three limitations persist: (i) most methods lack type-I error control for flagged anomalies; (ii) time series anomaly detectors Blázquez-García et al. (2021); Zamanzadeh Darban et al. (2024); Xu et al. (2022) can be adapted to unstructured texts via embeddings, but only assuming dense, regularly sampled streams; and (iii) these methods do not support SL designs with subject-specific, irregular observation times and evolving trajectories, hence missing on the individual level dynamic trends in the anomalies. Functional data analysis methods for SL anomaly detection exist (Sun & Genton, 2011; Dai & Genton, 2018; Hubert et al., 2015; Gervini, 2009), but they operate on structured numeric functions rather than unstructured text and likewise lack formal false-positive guarantees. As a result, there is no end-to-end solution that transforms SL texts into trajectory-aware feature representations and detects anomalies with explicit type-I error control.

**Our Contributions**   We introduce LLmFPCA-detect, a novel framework that combines LLM-based embeddings with sparse mFPCA to enable covariate-informed data segmentation and type-I error controlled anomaly detection in sparsely observed, longitudinal, heterogeneous text data, yielding feature representations suitable for incorporating SL texts in a wide range of downstream tasks. LLmFPCA-detect is broadly applicable to settings involving subjects with time-stamped text records that arrive irregularly over time. While we focus on sparsely sampled scenarios, the methodology can be readily adapted to densely observed data. We demonstrate the effectiveness of LLmFPCA-detect through its application to the Amazon Reviews dataset (Amazon data) and the Wikipedia talk-page comment streams (Wiki data). The key components of the framework, as illustrated in Figure 1, are:

1. **Representation** We derive domain-appropriate LLM embeddings for each time-stamped text. For the Amazon Reviews dataset, we embed the texts using emotion scores based on Plutchik's Wheel of EmotionsPlutchik (1980), which identifies eight primary emotions as the foundation for all

others. For the Wikipedia request–comment stream, we obtain toxicity and aggression scores using GPT for each comment to compare against findings from human-annotated scores.

2. **Learning trajectory representations and detection with guarantees** The numeric trajectories form multivariate SL data, which are processed using the mFPCA pipeline to obtain multivariate functional principal component (mFPC) scores. These scores, combined with baseline covariates, are used for covariate-informed clustering. Anomalies are then detected in an unsupervised manner by: i) screening points in the tails of the cluster-specific mFPC score distributions, and ii) statistically testing the screened points while controlling for multiple comparisons. The identified anomalies are further analyzed to localize time window specific deviations in the population.

3. **Interpretability and insights** We use LLMs to extract keywords from texts associated with each cluster and flagged window, revealing dynamic, human-interpretable signals that explain why the flagged discovery matters.

**Organization** The rest of the paper is organized as follows. Section 2 provides the motivation for the clustering and anomaly detection steps of LLmFPCA-detect. Section 3 outlines the methods, estimation procedures, and algorithms that make up the different steps in LLmFPCA-detect. Sections 4.1 and 4.2 demonstrates the application of LLmFPCA-detect to customer journey data from Amazon reviews and to Wikipedia request–comment streams, illustrating its cross-domain applicability. Additional details and experiments are provided in the Appendix.

## 2 MOTIVATION AND FRAMEWORK

**LLmFPCA-detect Pipeline**

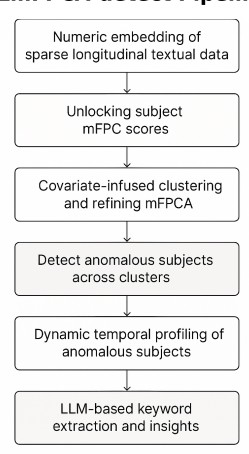

In this section, we present the foundational framework underlying LLmFPCA-detect.

**Multivariate functional data representation** For each subject $i = 1, \ldots, N$, a random function $\boldsymbol{X}_i \in L^2(\mathcal{T})^p$ is observed on a discrete, potentially irregular and sparse time grid $\{T_{ij}\}_{j,i=1}^{N_i,N}$ along with baseline covariates $\boldsymbol{Z}_i \in \mathbb{R}^q$ where $(\boldsymbol{X}, \boldsymbol{Z}) \sim \mathbb{P}$, with $\mathbb{P}$ being the joint distribution of $(\boldsymbol{X}, \boldsymbol{Z})$. The population mean function is defined as $\boldsymbol{\mu}(t) = \mathbb{E}(\boldsymbol{X}(t))$, and the covariance surface for $s, t \in \mathcal{T}$ is given by $\mathbb{C}(s,t) = \mathbb{E}\{(\boldsymbol{X}(s) - \boldsymbol{\mu}(s)) \otimes (\boldsymbol{X}(t) - \boldsymbol{\mu}(t))\}$ with entries $\mathbb{C}_{ij}(s,t) = \mathrm{Cov}(X^{(i)}(s), X^{(j)}(t))$ is assumed to satisfy the conditions of Proposition 2 in Happ & Greven (2018). Then, $\boldsymbol{X}$ admits a multivariate Karhunen–Loève expansion (Propositions 3 and 4 in Happ & Greven (2018))

$$\boldsymbol{X}(t) = \boldsymbol{\mu}(t) + \sum_{j=1}^{\infty} \rho_m \boldsymbol{\psi}_m(t)$$

Figure 1: Proposed framework

where $\rho_m = \langle \boldsymbol{X}(t) - \boldsymbol{\mu}(t), \boldsymbol{\psi}_m(t) \rangle$ with $\mathrm{Cov}(\rho_m, \rho_n) = \lambda_m \mathbb{I}\{m = n\}$, and $\lambda_1 \geq \lambda_2 \geq \cdots \geq 0$ are the eigenvalues of the covariance operator associated with $\mathbb{C}$. The corresponding eigenfunctions $\boldsymbol{\psi}_m, m \in \mathbb{N}$ serve as the multivariate functional principal components, with $\rho_m$ being the associated mFPC scores. If $\boldsymbol{X}$ admits a finite expansion with $M$ principal components, Proposition 5 in Happ & Greven (2018) establishes how mFPCA of $\boldsymbol{X}$ relates to univariate functional principal component analysis (uFPCA) of each component $X^{(d)}(\cdot) \in L^2(\mathcal{T})$ for $d = 1, \ldots, p$.

**Data heterogeneity and anomalies** Suppose the trajectories $\{\boldsymbol{X}_i\}_{i=1}^N$ belong to $K$ distinct clusters, denoted by $\mathcal{C}_1, \ldots, \mathcal{C}_K$, with $\bigcup_k \mathcal{C}_k = \{1, \ldots, n\}$ and $\mathcal{C}_k \cap \mathcal{C}_j = \emptyset$ for $j \neq k$. Observations in $\mathcal{C}_k$ are generated according to the distribution $\mathbb{P}_k$, yielding the overall mixture $\mathbb{P} = \sum_{k=1}^K \pi_k \mathbb{P}_k$ with $(\pi_1, \ldots, \pi_K)$ denoting cluster proportions. For $i \in \mathcal{C}_k$, assume that $\boldsymbol{X}_i$ admits a finite multivariate Karhunen–Loève expansion $\boldsymbol{X}_i(t) = \boldsymbol{\mu}_k(t) + \sum_{m=1}^M \rho_{im} \boldsymbol{\psi}_m(t), \quad i \in \mathcal{C}_k$, where $\boldsymbol{\mu}_k \in L^2(\mathcal{T})^p$ is the cluster-specific mean function, and $\boldsymbol{\psi}_m \in L^2(\mathcal{T})^p$ are shared eigenfunctions across clusters. To incorporate possible measurement errors and anomalies, we observe

$$\boldsymbol{Y}_i(t) = \boldsymbol{X}_i(t) + \boldsymbol{\eta}_i(t) + \boldsymbol{a}_i(t),$$

where $\boldsymbol{\eta}_i, \boldsymbol{a}_i \in L^2(\mathcal{T})^p$ capture the measurement errors and anomalies respectively. These are assumed to be jointly independent of $\boldsymbol{X}_i$, $i = 1, \ldots, n$, with $\mathbb{E}(\boldsymbol{\eta}_i(t)) \equiv 0$ for all $t \in \mathcal{T}$, and

$\mathrm{Cov}(\eta^{(j)}(s), \eta^{(k)}(t)) = \sigma_\eta^2 \mathbb{I}_{s=t}$ for all $j, k \in \{1, \ldots, p\}$. The term $\boldsymbol{a}_i \equiv \boldsymbol{0}$ almost surely for all $i \in \mathcal{A}_0^C$, where $\mathcal{A}_0 \subset \{1, \ldots, N\}$ denotes the set of anomalous subjects. For each $i \in \mathcal{A}_0$, we assume $\boldsymbol{a}_i(t) \neq \boldsymbol{0}$ for some $t \in \mathcal{T}_0 \subset \mathcal{T}$ almost surely. We employ trimmed $k$-means to recover the clusters accurately despite being contaminated with outliers; for details on cluster recovery see Section D.1 in Appendix D.

**Calibrating the anomalies** After the clusters are recovered, the anomalous observations in $\mathcal{A}_0$ are assigned to one of the clusters $\mathcal{C}_1, \ldots, \mathcal{C}_K$. To detect $\mathcal{A}_0$ in an unsupervised manner, we perform a screening step within each cluster by examining the tails of the FPC score distribution, approximating $\mathcal{C}_k \cap \mathcal{A}_0$ by $\mathcal{A}_0^{k,\epsilon} \subset \mathcal{C}_k$ (see Appendix E). The distribution of FPC scores in the clean subset $\mathcal{C}_k \cap \mathcal{A}_0^C$ is then used to recover $\mathcal{C}_k \cap \mathcal{A}_0$ with confidence. In practice, each cluster $\mathcal{C}_k$ is randomly split into two subsets, and the non-screened portion is used to calibrate the anomaly detection procedure; see Theorem E.1 in Appendix E for theoretical guarantees. Finally, based on the detected anomalous set $\mathcal{A}_0$, we analyze the corresponding keywords across different time windows.

The foregoing framework outlines a pipeline for obtaining cluster-specific feature representations and type-I controlled anomaly detection in fully observed multivariate functional trajectories with possible measurement errors. In SL settings, each subject is observed at random time points $T_{ij}$ for $i = 1, \ldots, N$, $j = 1, \ldots, N_i$, with $T_{ij} \in \mathcal{T}$. These time points $T_{i1}, \ldots, T_{iN_i}$ are assumed i.i.d. and independent of $\boldsymbol{X}_i$ and $\boldsymbol{\eta}_i$ for all $i$. The number of measurements $N_i$ is random, reflecting sparse and irregular designs, and $N_i$, for $i = 1, \ldots, N$, are assumed i.i.d. and independent of all other random elements. In practice, we observe $Y_i(T_{ij})$, $j = 1, \ldots, N_i$, $i = 1, \ldots, N$, and all relevant quantities must be estimated from these noisy observations. Section 3 outlines the estimation details and algorithms for this pipeline, including steps for incorporating the underlying textual data.

## 3 METHODS: PIPELINE AND ESTIMATION

**From SL Texts to Numeric Embeddings** The first step maps each time-stamped text $K_i(T_{ij})$ to a $p$-dimensional vector via a fixed embedding

$$\Phi : \mathcal{X} \longrightarrow \mathbb{R}^p, \qquad \boldsymbol{Y}_i(T_{ij}) = \Phi\big(K_i(T_{ij})\big), \tag{1}$$

where $\Phi$, is implemented via LLM prompting, held constant across subjects, and deterministic (the same text yields the same vector). For subject $i$ this yields the multivariate trajectory $\{\boldsymbol{Y}_i(T_{ij})\}_{j=1}^{N_i}$, whose coordinates are modeled jointly using mFPCA (e.g. Plutchik emotion embeddings for Amazon reviews; see Sections C and 4.1). Each subject also has baseline, time-invariant covariates $\boldsymbol{Z}_i \in \mathbb{R}^q$ (e.g. average rating, review length, engagement duration).

---

**Algorithm 1** Multivariate Functional Principal Component Analysis (mFPCA)

**Input:** SL data: $\{\boldsymbol{Y}_i(T_{ij})\}_{j=1}^{N_i}$ for $i = 1, \ldots, N$.

1: $(\{\hat{\xi}_{ik}^{(d)}\}_{i=1,k=1}^{N,K_d}, \hat{\mu}^{(d)}(t), \{\hat{\phi}_k^{(d)}(t)\}_{k=1}^{K_d}) \leftarrow \texttt{uFPCA}(\{(T_{ij}, Y_i^{(d)}(T_{ij}))\}_{i,j})$ for each dimension $d = 1, \ldots, p$. $\qquad \triangleright$ Algorithm 4; only scores are used below

2: $\hat{\boldsymbol{\Xi}}_i \leftarrow (\hat{\xi}_{i1}^{(1)}, \ldots, \hat{\xi}_{iK_1}^{(1)}, \ldots, \hat{\xi}_{i1}^{(p)}, \ldots, \hat{\xi}_{iK_p}^{(p)}), i = 1, \ldots, N \qquad \triangleright$ Stack univariate FPC scores

3: Define matrix $\hat{\boldsymbol{\Xi}} \in \mathbb{R}^{N \times M}$ with rows $\hat{\boldsymbol{\Xi}}_i$ where $M = \sum_{d=1}^p K_d$.

4: $\hat{\mathbf{C}}_\Xi \leftarrow \frac{1}{N-1} \hat{\boldsymbol{\Xi}}^\top \hat{\boldsymbol{\Xi}}$. $\qquad \triangleright$ Compute covariance matrix

5: Perform eigen-decomposition of $\hat{\mathbf{C}}_\Xi$ to obtain eigenvalues $\{\hat{\lambda}_m\}_{m=1}^M$ and eigenvectors $\{\hat{\boldsymbol{v}}_m\}_{m=1}^M$.

6: $\hat{\psi}_m^{(d)}(t) \leftarrow \sum_{k=1}^{K_d} \hat{v}_{m,k}^{(d)} \hat{\phi}_k^{(d)}(t), d = 1, \ldots, p$ and $m = 1, \ldots, M$. $\triangleright$ Multivariate eigenfunctions

7: $\hat{\rho}_{im} \leftarrow \hat{\boldsymbol{\Xi}}_i^\top \hat{\boldsymbol{v}}_m$ for $i = 1, \ldots, N$ and $m = 1, \ldots, M$ $\qquad \triangleright$ Compute mFPC scores

**Output:** Tuple of estimated mFPC scores, eigenfunctions and mean curves: $\{\hat{\rho}_{im}, \hat{\psi}_m, \hat{\boldsymbol{\mu}}\}_{i,m=1}^{N,M}$.

---

**Dynamic Trajectory Representations using mFPCA** Algorithm 1 details the estimation steps of the mFPCA setup outlined in Section 2. Starting from $\{\boldsymbol{Y}_i(T_{ij})\}_{j=1}^{N_i}$, we estimate the mFPC scores $\hat{\rho}_{im}$ by building on the univariate functional principal component analysis (uFPCA) of each $\{Y_i^{(d)}(T_{ij})\}_{j,i=1}^{N_i,N}$ for $d = 1, \ldots, p$. The algorithm follows the approach in Happ & Greven (2018),

using estimated quantities from uFPCA including the mean functions $\hat{\mu}^{(d)}(t)$, eigenfunctions $\hat{\phi}^{(d)}(t)$ and univariate FPC scores $\hat{\xi}_{ik}^{(d)}$; for details see Algorithm 4 in Section C.2 and Yao et al. (2005).

---

**Algorithm 2** Detecting anomalous subjects within a cluster $\hat{\mathcal{C}}$

---

**Input:** Subject cluster $\hat{\mathcal{C}}$; data $\{\boldsymbol{Y}_i(T_{ij}) : i \in \hat{\mathcal{C}}\}$; significance levels $\alpha_1, \alpha$ (where $\alpha_1 > \alpha$).

1: Obtain mFPC scores $\{\hat{\rho}_{im}^{\hat{\mathcal{C}}}\}_{i \in \hat{\mathcal{C}}, m=1,\ldots,B}$ corresponding to the top $B$ cluster-specific mFPC components using Algorithm 6 applied to $\{\boldsymbol{Y}_i(T_{ij}) : i \in \hat{\mathcal{C}}\}$.
         $\triangleright$ $B$: number of top mFPC components based on prop. of variance explained
2: Randomly partition $\hat{\mathcal{C}}$ into disjoint sets $I_1, I_2$ of equal size.
3: $(G_1, G_1^c) \leftarrow \texttt{ScreenPotentialOutliers}(I_1, \{\hat{\rho}_{im}^{\hat{\mathcal{C}}} : j \in I_1\}, B, \alpha_1)$.   $\triangleright$ Algorithm 7
4: $(G_2, G_2^c) \leftarrow \texttt{ScreenPotentialOutliers}(I_2, \{\hat{\rho}_{im}^{\hat{\mathcal{C}}} : j \in I_2\}, B, \alpha_1)$.
5: Initialize $\mathcal{A}^{(1)} \leftarrow \emptyset$.        $\triangleright$ Set of confirmed outliers for cluster $\hat{\mathcal{C}}$
6: $\mathcal{A}_{G_1}^{(1)} \leftarrow \texttt{ConfirmAnomalies}(G_1, G_2^c, \{\hat{\rho}_{im}^{\hat{\mathcal{C}}} : j \in G_1 \cup G_2^c\}, B, \alpha)$.    $\triangleright$ Algorithm 8
7: $\mathcal{A}_{G_2}^{(1)} \leftarrow \texttt{ConfirmAnomalies}(G_2, G_1^c, \{\hat{\rho}_{im}^{\hat{\mathcal{C}}} : j \in G_2 \cup G_1^c\}, B, \alpha)$.
8: $\mathcal{A}^{(1)} \leftarrow \mathcal{A}_{G_1}^{(1)} \cup \mathcal{A}_{G_2}^{(1)}$.

**Output:** Set of confirmed anomalous subjects $\mathcal{A}^{(1)} = \{(i, S_i) : i \in \hat{\mathcal{C}} \text{ is an outlier}, S_i \neq \emptyset\}$.

---

**Clustering and Anomaly Detection using mFPC Scores and Covariates** We segment subjects by clustering their estimated mFPC scores jointly with static covariates (Algorithm 5, Appendix D). For each estimated cluster $\hat{\mathcal{C}}_k$ we re-fit mFPCA using only its members (Algorithm 1; Algorithm 6), yielding cluster-specific means $\hat{\boldsymbol{\mu}}_k(t)$, eigenfunctions $\hat{\boldsymbol{\psi}}_m^k(t)$, updated scores $\hat{\rho}_{im}^k$ and and reconstructed trajectories (Equation equation 8).

---

**Algorithm 3** Dynamic temporal profiling of anomalous subjects

---

**Input:** Type 1 anomalies $\mathcal{A}^{(1)}$ (from Alg. 2 for cluster $\hat{\mathcal{C}}$); data $\{\boldsymbol{Y}_j(T_{jk}) : j \in \hat{\mathcal{C}}\}$; cluster means $\{\hat{\mu}_{\hat{\mathcal{C}}}^{(d)}(t)\}$ (from Alg. 6); Clean held-out sets $G_1^c, G_2^c$ & split info $I_1, I_2$ for $\hat{\mathcal{C}}$ (from Alg. 2); time windows $\{(a_w, b_w)\}_{w=1}^W$; significance level $\alpha$.

1: $(\{\bar{\boldsymbol{\mu}}_{\hat{\mathcal{C}}}^{(w)}\}_{w=1}^W, \{D_j^{(w)}\}_{j \in G_1^c \cup G_2^c, w=1,\ldots,W}) \leftarrow \texttt{ComputeWindowDeviations}(\{\boldsymbol{Y}_j(T_{jk}) : j \in G_1^c \cup G_2^c\}, \{\hat{\mu}_{\hat{\mathcal{C}}}^{(d)}(t)\}, \{(a_w, b_w)\}_{w=1}^W)$.   $\triangleright$ Compute scores for clean held-out set Alg. 9

2: $\mathcal{A}^{(2)} \leftarrow \texttt{IdentifyAnomalousWindows}(\mathcal{A}^{(1)}, \{\boldsymbol{Y}_i(T_{ij}) : i \text{ s.t. } (i, \_) \in \mathcal{A}^{(1)}\}$,
    $\{\bar{\boldsymbol{\mu}}_{\hat{\mathcal{C}}}^{(w)}\}, \{D_j^{(w)}\}, I_1, I_2, G_1^c, G_2^c$,
    $\{(a_w, b_w)\}_{w=1}^W, \alpha)$.    $\triangleright$ Identify anomalous windows for subjects (Alg. 10)

**Output:** Set of subject-indexed anomalous temporal windows $\mathcal{A}^{(2)} = \{(i, \mathcal{W}_i) : i \in \mathcal{A}^{(1)}, \mathcal{W}_i \neq \emptyset\}$.

---

Globally anomalous subjects will still be assigned to one of the $K$ clusters unless explicitly screened—a difficult task in heterogeneous data. To detect such cases post-assignment, we apply Algorithm 2 (with Algorithms 7 and 8; Appendix E). The procedure tests whether a subject's multivariate FPC scores deviate from the typical pattern of its assigned cluster $\hat{\mathcal{C}}$, using sample splitting and data-driven calibration to control multiplicity across principal components. It outputs flagged subjects $\mathcal{A}^{(1)} = (i, S_i)$, where $S_i$ records the outlying FPC directions—information that then guides localized anomaly analysis (Algorithm 3).

Subjects flagged by Algorithm 2 (set $\mathcal{A}^{(1)}$) may be anomalous only over portions of their trajectories. Algorithm 3 localizes these periods by comparing each subject's raw segments to the cluster mean, with data-driven calibration (Algorithm 9); implementation details are in Appendix E (Algorithms 9, 10). The output is $\mathcal{A}^{(2)} = (i, \mathcal{W}_i)$, where $\mathcal{W}_i$ denotes the time windows in which subject $i$'s trajectory departs from a clean cohort within that window. This step pinpoints atypical intervals and enables per-window anomaly flags, which feed into the final dynamic keyword profiling stage.

**Dynamic Keyword Profiling**   Finally, we describe intent extraction from anomalous reviews. For each subject $i$, let $S_i$, be the anomalous reviews. Challenges include lexical variation for similar semantics, shared stylistic drift across users, and scalability for large number of anomalous reviews. We maintain a time-ordered intent list $I_i^{(t-)}$ from reviews before time $t$. At time $t$, an LLM receives $I_i^{(t-)}$, top global intents observed before $t$, and the current review, and either matches an existing intent or proposes a new one. Full details appear in Algorithm 11 (Appendix F).

## 4   REAL DATA APPLICATIONS

### 4.1   MODELING DYNAMIC EMOTIONS IN AMAZON CUSTOMER REVIEWS

We use the Amazon Reviews corpus Hou et al. (2024), which includes 1,946 users and 22,032 reviews over five years, focusing on Automobile for the main analysis; Beauty & Personal Care and Sports & Outdoors supply user-level covariates (e.g., cross-category purchase share). Each review includes a user ID, timestamp, product title, text, and a 1–5 rating, with users posting over multiple years.

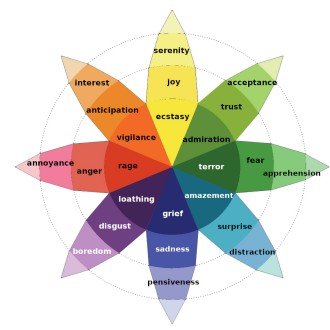

Figure 2: Plutchick's wheel of emotions

**Emotion embedding for text transcripts**   Plutchik's wheel of emotions provides a structured framework for mapping emotional states along opposing pairs, capturing both intensity and polarity (see Fig. 2 Semeraro et al. (2021)). We convert each transcript into four real-valued scores—joy–sadness, trust–disgust, fear–anger, and surprise–anticipation—on a continuous $[-1, 1]$ scale, where $-1$ and $1$ denote the extremes of each pole (e.g., grief vs. ecstasy), and intermediate values encode moderate intensity. A zero-shot GPT-3.5-Turbo prompt returns one scalar per axis (details and validation in Section C.1). Stacking these over time yields a 4-D timestamped embedding per subject, which serves as the input to LLmFPCA-detect for mFPCA and the subsequent steps.

| Rating | Review | Joy–Sadness | Trust–Disgust | Fear–Anger | Surprise–Anticipation |
|---|---|---|---|---|---|
| 5 | I use this great oil in all of my 150cc Scooters (was told to by a Scooter mechanic) and I've never had an engine problem. But this price is thru the roof, $17.50 for a single quart is STUPID...wally world sells it for $4.99...but its kinda funny that all of Amazon's oils are priced thru the roof | -0.8 | -0.6 | -0.8 | **-1** |
| 1 | Received this today and went to put it on my 3/8 extension for an oil filter change. The machining is pretty, but measurements are so poor I cannot get it on the extension to use. Absolute junk! I should have paid more attention to the negative review. | -0.77 | **-0.75** | -0.5 | -0.7 |

Table 1: Amazon customer reviews with emotion scores across four Plutchik dimensions.

Table 1 illustrates how emotion embeddings reveal customer pain points that are not captured by 5-star ratings alone. In the first example, a 5-star review shows strong sadness (–0.8), disgust (–0.6), anger (–0.8), and surprise (–1), indicating frustration with pricing despite overall satisfaction. The third example, also rated 1 star, shows high sadness (–0.77), disgust (–0.75), and surprise (–0.7), pointing to severe frustration over usability issues.

**Emotion mFPCA scores (Algorithm 1) improve predictive power over product ratings** We test whether review text improves forecasting of adverse outcomes (e.g., sudden rating drops) in Amazon Reviews. A "rating drop" is defined as the extreme percentile of each user's maximum gap between consecutive ratings. We compare two optimally tuned random-forest models on a class-balanced sample with identical baseline covariates—cluster labels from Algorithms 1–5 and purchase mix across categories. Model A summarizes past behavior by the mean Automobile rating; Model B replaces that single aggregate with emotion mFPC scores, capturing time-varying textual signals. On the test set, Model A: accuracy 0.542, precision 0.538, recall 0.596, F1 0.565, ROC–AUC 0.534. Model B improves all metrics—accuracy 0.609 (**+12.4%**), precision 0.610 (**+13.4%**), recall 0.603 (**+1.2%**), F1 0.606 (**+7.3%**), ROC–AUC 0.645 (**+20.8%**)–showing that compact emotion-trajectory features capture predictive signal beyond coarse star-rating averages.

**Clustering dynamics and case studies** Figure 3 plots mean emotion trajectories for the three clusters from purchasing proportions in Automobiles, Beauty & Personal Care, and Sports & Outdoors).

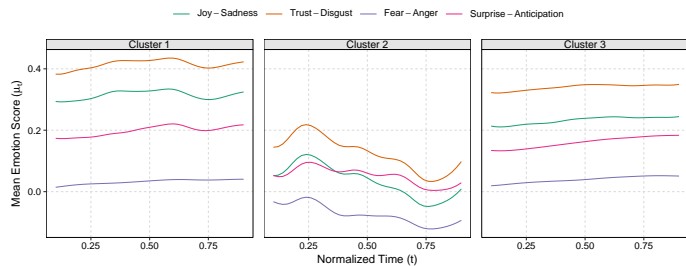

Figure 3: Mean emotion trajectories across the three user clusters. Curves represent mean scores for the Joy–Sadness, Trust–Disgust, Fear–Anger, and Surprise–Anticipation emotion dimensions.

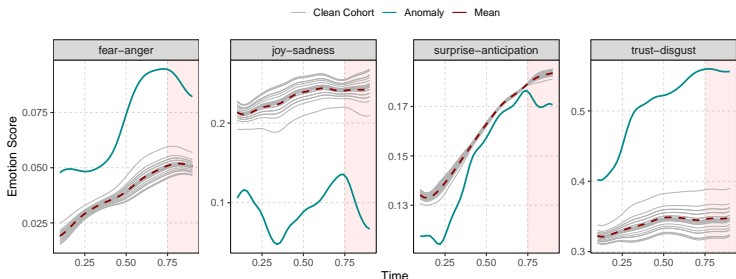

Figure 4: Mode of variation plot for a user along the fourth FPC (outlying) from cluster 3

Cluster 1 has the highest baseline across emotions–consistently stronger affect. Cluster 3 follows a similar temporal shape but is uniformly lower (milder affect). Cluster 2 departs most, with elevated sadness and anger, indicating sharper pain points. Because anomalies are scored relative to each cluster's mean, even upward shifts in positive emotion within Cluster 2 can register as anomalous. Section D.3 of the Appendix reports bootstrap analysis confirming cluster stability.

Through mode-of-variation plots (see Section C.2 for details) and corresponding review excerpts in the flagged time window, we show that the detected anomalies capture customer pain points. Figure 4 shows a user's emotional trajectory relative to Cluster 3. The user's emotions are consistently shifted from the cluster mean along the fourth eigenfunction in Cluster 3, with a pronounced spike in the fear–anger petal and a sharp drop in joy–sadness during the final time window—signaling a clear pain point. Review texts from this period reveal issues with mismatched parts, specifically a replacement door-handle cover with incorrect keyhole cut-outs. The dominant complaints relate to product fit and quality control. These insights suggest actionable interventions, such as enforcing compatibility checks at purchase and improving final-stage quality control by the seller.

Figure 5 shows a user exhibiting a dip–recovery emotional pattern along the second eigenfunction. Early in the timeline, all four emotion petals remain well below the cluster baseline. During the anomalous time window, there is a sharp rise in fear and surprise, driven by issues related to poor product quality. The user expresses frustration and regret, suggesting loss of brand trust. Key pain points include the failure of a critical component and confusion caused by missing documentation.

**Keyword profiling** After detecting anomalies, we perform keyword profiling (Algorithm 11 in Section F of the Appendix) to each flagged instance. Table 2 summarizes the keywords associated with anomalous points in each cluster. A quick glance shows that users in Cluster 2 tend to express broadly negative emotions, while Cluster 3 highlights more specific issues—such as missing cables and poor documentation—reflecting the more descriptive and varied nature of reviews in that group. Table 6 illustrates dynamic profiling of keywords; see Section F in the Appendix for details.

### 4.2 TRACKING TOXICITY AND AGGRESSION IN WIKIPEDIA REQUEST–COMMENT STREAM

We evaluate LLmFPCA-detect on the English Wikipedia request–comment stream to demonstrate cross-domain applicability. For each comment, we record the text, timestamp, structured user

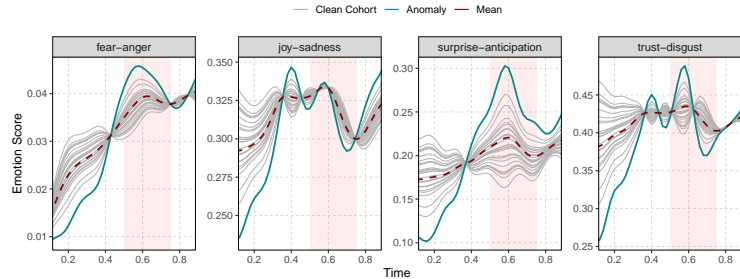

Figure 5: Mode of variation plot for a user along the second FPC (outlying) from cluster 1

| Cluster | Keywords |
|---------|----------|
| Cluster 1 | as described, good quality, perfect product, poor value for money, wrong size, poor fit |
| Cluster 2 | poor quality, poor value for money |
| Cluster 3 | as described, bulky design, good quality, good value for money, good design, leaks fuel, missing cable, quantity issue, poor documentation, wrong size |

Table 2: Group-level pain points detected across clusters

covariates, and crowdsourced ground-truth toxicity/aggression scores. This corpus exemplifies sparse longitudinal text: users post at irregular, infrequent intervals. The dataset was collected via the Wikipedia API, restricted to the user–talk and article–talk namespaces, and sourced from Wiki data. We retain comments from 2010–2015 authored by 925 pseudonymized users.

| Method | TW1 | TW2 | TW3 | TW4 | TW5 |
|--------|-----|-----|-----|-----|-----|
| LLmFPCA-detect (gpt-4o-mini) | **0.58** | **0.58** | **0.46** | **0.37** | **0.32** |
| Isolation Forest (BERT) | 0.41 | 0.33 | 0.25 | 0.23 | 0.39 |
| Isolation Forest (gpt-4o-mini) | 0.41 | 0.33 | 0.25 | 0.23 | 0.39 |

Table 3: F1 scores for anomalies detected by LLMFPCA-DETECT versus ground truth, compared with Isolation Forest on GPT-derived scores and a BERT baseline (segregated by time windows).

**Comparison with state-of-the-art** We assess anomaly detection on Wikipedia by treating human-annotated toxicity/aggression as surrogate ground truth and extracting GPT-derived toxicity/aggression scores from text via prompts. As a content-agnostic baseline, we use BERT embeddings (no explicit toxicity cues). We partition the timeline into five windows and, within each, define pseudo–ground-truth anomalies using Isolation Forest on the human scores plus user covariates (comment count, median inter-comment gap). We then run Isolation Forest on (i) GPT-derived scores and (ii) BERT embeddings (each with the same covariates) as baselines. Finally, we apply LLmFPCA-detect to the GPT-derived trajectories with the same covariates to flag anomalies across the five windows and compare against these baselines (Table 3).

**Cluster dynamics** LLmFPCA-detect flags not only one-off vandalism or brief flare-ups by otherwise well-behaved contributors, but also sustained problematic behavior and its mode of deviation. For example, Cluster 1 outliers tend to post unusually high volumes or engage in extended policy disputes, whereas Cluster 2 outliers show short, intense bursts of toxic language. Table 4 presents representative cases with brief excerpts and the corresponding anomalous time window. In Cluster 1, the dominant pattern is procedural friction—disagreements about process (e.g., whether a proposed mentorship program requires further consensus) rather than direct attacks. By contrast, Cluster 2 features overt hostility, where procedural disagreements escalate into personal or confrontational language. Additionally, Appendix D.3 reports bootstrap analyses confirming stability of the obtained clusters.

**Keyword profiling** Dynamic keyword profiling makes each anomaly interpretable (Table 5). Rather than an opaque outlier score, moderators see the top terms that triggered the flag, revealing the

| Cluster | User ID | Comment excerpt (abridged) | Label |
|---|---|---|---|
| 1 | 10783082 | "…If that's how you want it. I will talk to this to ANI if necessary …" | 1 |
| 1 | 10756369 | "=== Adopt Me === Here is a proposal for a new mentorship process …" | 1 |
| 2 | 2305952 | "OK, maybe I was wrong. I'm sorry, but don't try me again …" | 5 |
| 2 | 2305952 | "No, that's irrelevant. Your source is garbage, stop spamming it." | 5 |

Table 4: Examples from the Wikipedia comment stream where detected anomalies match crowd-sourced annotations, showing cluster ID, anonymized user ID, excerpt, and toxicity/aggression label.

| Cluster 1 (Window) | Top keywords (LLmFPCA-detect) | Theme |
|---|---|---|
| W1 | consensus, policy, "WP: ANI" | Policy enforcement friction |
| W2 | civility, manners, please, courtesy | Soft-skills reminders |
| W3 | backlog, deadline, stall, formalise | Procedural urgency |

| Cluster 2 (Window) | Top keywords (LLmFPCA-detect) | Theme |
|---|---|---|
| W4 | nonsense, garbage-source, stop-spamming | Direct hostility |
| W5 | revert, vandal, warning, block, "3RR" | Conflict over content |
| W5 | wasting-time, already-explained | Moderator fatigue |

Table 5: Dynamic–keyword profiling makes each anomaly legible. In this Wikipedia setting, instead of an opaque outlier score, the moderator sees the top keywords that drove the statistical flag.

concerns underlying anomalous behavior. In this corpus, Cluster 1 anomalies are predominantly procedural—e.g., disputes over which venue (WP:ANI, etc.) should adjudicate. Cluster 2, by contrast, exhibits explicit antagonism: personal attacks, contempt for sources ("garbage-source"), and edit-war jargon. The exasperation lexicon ("wasting time," "already explained") further signals moderator fatigue—an operational risk that steady-state toxicity metrics would miss.

Identifying peak–hostility windows (e.g., Window 5) with LLmFPCA-detect enables proactive moderation, such as temporarily throttling edits. In Cluster 1, the dominant issue is procedural friction, suggesting policy fixes like clearer closure rules or targeted sanctions. Keyword profiling pinpoints specific, time-bounded situations where light-touch actions can prevent rule violations and burnout. Linking time windows to salient terms reveals root causes and supports proportionate, domain-specific responses instead of one-size-fits-all bans.

## 5 CONCLUSION

LLmFPCA-detect provides an end-to-end framework for sparse longitudinal (SL) text by integrating LLM-embeddings with functional data analysis. LLmFPCA-detect tackles key challenges in such datasets—including sparsity, irregularity, noise, and semantic complexity—by embedding text into meaningful numeric representations, followed by mFPCA which is used for user segmentation, anomaly detection, and dynamic intent profiling across large SL text datasets, a setting that remains largely unaddressed in the literature. Applied to Amazon customer reviews, LLmFPCA-detect successfully uncovers emotion dynamics and identifies critical pain points in the customer journey, offering valuable insights for consumer analytics. We demonstrate the utility of LLmFPCA-detect on English Wikipedia request–comment stream to detect toxic comments, where the detected anomalies align well with crowdsourced human annotations. The flexibility of LLmFPCA-detect makes it applicable to other domains such as healthcare, education, and social media where SL text data is routine. Future work includes establishing theoretical guarantees based on mFPCA estimates rather than fully observed trajectories, and extending LLmFPCA-detect to other supervised and unsupervised tasks on SL text datasets.

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
