# OpenReview forum: "LLmFPCA-detect: LLM-powered Multivariate Functional PCA for Anomaly Detection in Sparse Longitudinal Texts"
_ICLR.cc/2026/Conference — Submitted to ICLR 2026_

### Official Review · Reviewer_hBza · 2025-10-30

**Soundness:** 3
**Presentation:** 2
**Contribution:** 2
**Rating:** 4
**Confidence:** 4

**Summary:**

This paper presents an approach on clustering and finding anomalies in sparse longitudinal data: data with irregular timestamps associated to one user. It combines LLM generated embeddings with mFPCA for clustering and anomaly detection. The authors argue that current methods either ignore temporal structure or miss guarantees for errors. The method is evaluated on two datasets and some insights are extracted from the clusters, based on the most frequent keywords.

**Strengths:**

- Timely problem: Anomalies in SL text is a relevant topic that has applications in several fields.

- Clear explanations and intuitions at the high level: In spite of the heavy formalizations, one can follow easily the ideas behind the paper. The high-level view of the pipeline is clear.

- It seems that adding the emotion mFPC scores improves predictions on some downstream tasks.

**Weaknesses:**

- Formalization is not very rigorous. Some concepts are not properly explained, which makes the reading hard. For example, Algorithm 2 uses Algorithm 7 and 8, but they are not explained. These are important components of the overall pipeline, so at least an intuition or a high-level formalization would improve the quality of the paper.
- The scientific contribution is not very novel. mFPCA is well established and appending the LLM embeddings plus Algorithms 2 and 3, which only apply some postprocessing to the output of mFPCA is not an idea that one can then transfer to other domains.
- Unclear explanations of experiments, see below for questions. There are some things that are not clear about the experiments, so it is hard to assess their quality.

**Questions:**

**Data heterogeneity and anomalies**

- If the Karhunen-Loeve expansion is applied as written, clusters differ only in the cluster mean. Why is it valid to assume all clusters share the same models of variation (eigenfunctions)? In the Amazon example, should we expect the "poor-quality reviews" cluster to share eigenfunctions with the "good-reviews" cluster?
- Line 164; What is T_0? Also, if anomalies are defined by $a_i(t) \neq 0$, how do you distinguish a true anomaly from measurement error when $a_i(t)$ is small?
- Line 170: You approximate $C_k \cup A_0$ by $A_0^{k, \epsilon}$, but later use the distribution of FPC scores from the "clean" subset $C_k \cup A^C_0$. How is that clean distribution obtained if you only had an approximation earlier?

**Methods: Pipeline and estimation**

- Line 237: What does "clustering their estimated mFPC scores jointly with static covariates" mean operationally? Are covariates appended as additional dimensions in the clustering vectors? If so, how are covariates encoded and scaled with respect to the mFPC scores?
- Algorithm 3: How are detection windows defined (length, stride)? How many windows are used per subject? Are they overlapping or disjoint, and how sensitive are results to this choice?
- The Dynamic Keyword Profiling description is quite short. How exactly are LLMs used for this? What text is passed to the model and how are outputs refined?

**Experiments**

- How do you choose the embedding function in practice? For Amazon, how were the LLM-derived emotion scores obtained? Why use zero-shot rather than few-shot prompts?
- Why was an older GPT-3.5 model used? Do you compare embeddings or labelers from other models and report performance differences?
- Line 373: What is Table 6? (It seems referenced but not present)
- How many anomalies were detected in total, and how are they distributed across clusters and time windows?
- On Wikipedia, you mention BERT baselines, but why isn't there a comparison of LLmFPCA-detect using BERT-based encoders versus GPT-based encoders?
- How were comments selected from the full Wikipedia dataset? For the 925 users, how many total comments and time points are included after filtering?
- Which GPT model produced toxicity/aggression scores, and how do these scores correlate with human annotations?

---

> ### Author Response · Authors · 2025-11-26
> **Response to Anon. Reviewer hBza (Part 1/6)**
>
> We sincerely thank you for the detailed and thoughtful review, as well as for your questions, comments, and feedback, which we believe will substantially improve the manuscript. Below, we present our rebuttal, addressing the questions first, followed by the identified weaknesses in their original order, and we would be grateful for the opportunity to respond to any additional questions or concerns during the discussion period.
>
> (continued below)

---

> ### Author Response · Authors · 2025-11-27
> **Response to Anon. Reviewer hBza (Part 2/6)**
>
> ## Questions
>
> **Q1**
>
> Thank you for raising this important point. We agree that, in general, different behavioral clusters (e.g., “poor-quality’’ vs. “good-reviews’’ users) may exhibit distinct internal modes of variation. Our methodology is explicitly designed to accommodate such heterogeneity through a two-stage “coarse-to-fine’’ procedure, where the assumption of shared eigenfunctions is used only for initialization—not as the final modeling assumption.
>
> * **Why shared eigenfunctions appear in the theoretical setup:** In the unsupervised setting, we encounter a fundamental “chicken-and-egg’’ problem: cluster-specific eigenfunctions cannot be estimated before the clusters themselves are known. Therefore, in Stage 1 (Section 2), we apply FPCA to the entire cohort and project all subjects onto a common latent score space obtained from the globally estimated eigenfunctions. This shared-basis assumption is introduced purely to make the theory tractable, and our framework—and even the theory—can in principle be extended to heterogeneous eigenfunctions. Since doing so would substantially complicate notation and analysis, and our focus is methodological, we have chosed to adopt the shared eigenfunction assumption for simplicity.
> * **How the methodology accommodates heterogeneous eigenfunctions:** Once the clusters are identified, as detailed in **Algorithm 6** (Cluster-Specific mFPCA), we re-run the estimation pipeline separately within each cluster $\mathcal{C}_k$. This yields cluster-specific eigenfunctions $\hat{\psi}_m^k(t)$ and means $\hat{\mu}_k(t)$, which better capture group-specific dynamics, reduce reconstruction error for the “clean’’ cohort, and substantially improve the sensitivity of the anomaly detection steps (Algorithms 2–3). Thus, while shared eigenfunctions serve as an initial device for segmentation, the final modeling is fully heterogeneous across clusters.
> * **Application to Amazon Reviews:** In the final Amazon analysis, the “poor-quality’’ and “good-reviews’’ clusters are not forced to share eigenfunctions. After segmentation, each cluster is modeled with its own mFPCA decomposition. This allows the framework to capture qualitatively different modes of variation: for instance, volatility in “Anger’’ or “Disgust’’ may dominate the negative-review cluster, whereas the positive-review cluster may vary primarily along dimensions such as “Joy’’ or “Trust.’’ The reconstructed trajectories (Eq. 8) reflect these distinct dynamics.
>
> The shared eigenfunction assumption is used only for the initial unsupervised segmentation and only for theoretical simplicity. The final LLmFPCA-detect model is fully capable of—and explicitly designed for—cluster-specific eigenfunctions, which is precisely what we implement in the empirical applications.
>
> **Q2**
>
> * **Definition of $\mathcal{T}_0$:** $\mathcal{T}_0$ denotes the subset of the time domain on which the anomaly component $a_i(t)$ is active. As specified in Section 3, anomalous subjects satisfy $a_i(t)\neq 0$ for some $t\in\mathcal{T}_0$, while non-anomalous subjects have $a_i\equiv 0$. This allows the model to capture transient rather than persistent deviations.
> * **Distinguishing Anomalies from Measurement Error:** To distinguish anomalies from measurement error, we note that (i) measurement error $\eta_i(t)$ is modeled as mean-zero white noise and is attenuated by the mFPCA smoothing step, whereas structural deviations $a_i(t)$ survive reconstruction; (ii) Appendix E.1 assumes the anomalous projection has support bounded away from zero ($a_{\min}>0$), ensuring detectability; and (iii) Algorithm 7 screens only the extreme tails of the cluster-specific mFPC score distribution, so small noise-induced deviations remain in the “clean'' cohort. Together, these ingredients ensure that measurement error and anomalies are cleanly separated in both the model and the detection pipeline.
>
> (continued below)

---

> ### Author Response · Authors · 2025-11-27
> **Response to Anon. Reviewer hBza (Part 3/6)**
>
> **Q3**
>
> Thanks for raising this question; we appreciate the opportunity to clarify. We approximate the anomalous set $\mathcal{C}_k \cap \mathcal{A}_0$ using the screened set $\mathcal{A}_0^{k,\epsilon}$. Theorem E.1 shows that, for an appropriately chosen $\epsilon$, this approximation is conservative, in the sense that $\mathcal{C}_k \cap \mathcal{A}_0 \subset \mathcal{A}_0^{k,\epsilon}$.
>
> To estimate the distribution of the FPC scores for the clean subset $\mathcal{C}_k \cap \mathcal{A}_0^c$, we use the complement $(\mathcal{A}_0^{k,\epsilon})^c$. Under the above guarantee, we have $(\mathcal{A}_0^{k,\epsilon})^c \subset \mathcal{C}_k \cap \mathcal{A}_0^c$, so the FPC score distributions over these two sets coincide. When the proportion of anomalies within each cluster is not large—a standard and reasonable assumption—$(\mathcal{A}_0^{k,\epsilon})^c$ is a good approximation of the clean subset, and the empirical FPC score distribution over $(\mathcal{A}_0^{k,\epsilon})^c$ accurately reflects that of $\mathcal{C}_k \cap \mathcal{A}_0^c$.
>
> **Q4**
>
> Yes, operationally, the static covariates are appended as additional dimensions to the clustering vectors. As detailed in **Algorithm 5** (Appendix D.2), the procedure is as follows:
> * **Concatenation:** For each subject $i$, we construct a joint feature vector $\mathbf{w}_i \in \mathbb{R}^{M+q}$ by concatenating the vector of estimated mFPC scores $\hat{\boldsymbol{\rho}}_i \in \mathbb{R}^M$ with the vector of static covariates $\boldsymbol{Z}_i \in \mathbb{R}^q$.
> * **Scaling (Whitening):** To ensure that the mFPC scores and static covariates—which likely possess different units and variances—contribute equally to the clustering distance metric, we perform a **whitening** transformation. We compute the empirical covariance matrix $\hat{\Sigma}_w$ of the joint vectors $\mathbf{w}_i$ and multiply the centered vectors by $\hat{\Sigma}_w^{-1/2}$. This results in a transformed feature set $\{\tilde{\mathbf{w}}_i\}$ where all dimensions have zero mean and identity covariance, effectively normalizing the scale and removing correlation between the functional scores and the static metadata before K-means is applied.
>
> **Q5**
>
> These are all great points. Windows are defined as a set of $W$ contiguous, non-overlapping time intervals $\{(a_w, b_w]\}_{w=1}^W$ that partition the observation domain $\mathcal{T}$. We employ disjoint partitions rather than sliding windows to facilitate independent statistical testing (Bonferroni-corrected) within each segment.
> * **Experimental Settings:**
>     * For the **Amazon Reviews** application, we partitioned the customer journey into $W=4$ sequential windows.
>     * For the **Wikipedia** application, we partitioned the comment stream into $W=5$ windows.
> * **Sensitivity:** We explicitly evaluated the impact of window-based detection in our Simulation Study (Appendix B.2). We found that for subjects with sustained deviations (Clusters 1 and 3), detection is robust ($>0.8$ recall). However, for subjects with subtler, localized anomalies (Cluster 2), detection is sensitive to the windowing resolution. This finding serves as the primary justification for Algorithm 3: global screening alone may dilute transient signals, whereas targeted window testing recovers these “fragile’’ anomalies that are otherwise missed.
>
> To summarize, the choice of the number and length of windows is application-specific and depends on the sparsity of the observation times. If too many windows are used, many will contain too few observations to support reliable inference, and the appropriate tradeoff therefore varies by setting. In practice, one may also incorporate domain knowledge and test only a subset of windows—for example, those that are most relevant for a given anomalous subject—rather than testing all windows uniformly.
>
> (continued below)

---

> ### Author Response · Authors · 2025-11-27
> **Response to Anon. Reviewer hBza (Part 4/6)**
>
> **Q6**
>
> As detailed in **Algorithm 11** (Appendix F), the LLM is not merely used for summarization but functions as a **semantic classifier and standardized label generator** within a dynamic feedback loop.
> The process operates as follows for each anomalous review $r_j^{(t)}$ at time $t$:
> * **Input Context:** The LLM is prompted with a three-part context tuple $\{I_j^{(t-)}, \mathcal{K}_{-j}^{(t-)}, r_j^{(t)}\}$:
>     1. **User History ($I_j^{(t-)}$):** The list of intent keywords previously identified for this specific user (e.g., “shipping delay’’).
>     2. **Global Context ($\mathcal{K}_{-j}^{(t-)}$):** The top-$k$ most frequent intent keywords currently active across the *entire* population (e.g., “poor quality,’’ “wrong size’’).
>     3. **Current Text ($r_j^{(t)}$):** The raw text of the review flagged as anomalous.
> * **Classification Logic (Matching vs. Generation):** The LLM is instructed to perform a semantic matching task first. It evaluates whether the current review $r_j^{(t)}$ semantically aligns with any existing intent in the User History or Global Context lists.
>     * **If a match is found:** The review is assigned that standardized label (e.g., mapping “The box was crushed’’ to the existing label “damaged packaging’’). This standardization is crucial for aggregating consistent statistics across users.
>     * **If no match is found:** The LLM generates a *new*, concise intent phrase ($i_{new}$) that captures the novel issue (e.g., “missing keyhole cutout’’). This new label is then added to the user's history $I_j^{(t)}$ for future comparisons.
> * **Refinement:** This dynamic update ensures that the intent vocabulary evolves over time. The system learns to recognize new emerging anomalies (e.g., a manufacturing defect in a specific batch) while maintaining consistency for recurring issues, preventing the “lexical variation" problem where similar issues get fragmented into different synonyms.
>
> **Q7**
>
> * **Choice of Embedding Function ($\Phi$):** The choice of $\Phi$ is domain-dependent and guided by the specific analytical goal. The framework requires $\Phi$ to map unstructured text into a meaningful numeric space that captures the latent variation relevant to the user's objective.
>     * For **Amazon Reviews**, we selected **Plutchik's Wheel of Emotions** because standard sentiment analysis (positive/negative) fails to distinguish between distinct pain points like “Anger’’ (active hostility) versus “Sadness’’ (passive disappointment).
>     * For **Wikipedia**, we selected **Toxicity and Aggression** scores because the goal was to identify moderation events and harassment.
> * **Obtaining Amazon Scores:** We employed **GPT-3.5-Turbo** via a prompting pipeline with a fixed seed. Elaborate details are present in Section C of the appendix.
>     * We mapped the text (Review Title + Body) to four continuous axes corresponding to Plutchik's opposing pairs (e.g., Joy–Sadness) on a scale of $[-1, 1]$.
>     * The scoring was guided by a strict rubric where specific intensity levels mapped to numeric ranges (e.g., “Serenity’’ $\to (0, 1/3]$, “Ecstasy’’ $\to (2/3, 1]$).
> * **Zero-Shot vs. Few-Shot:** We deliberately chose a **zero-shot** approach with a detailed instructional rubric.
>     * **Rubric Precision:** As detailed in Appendix C.1, the prompt explicitly defines the numeric intervals for each emotion intensity (e.g., “Grief’’ is $[-1, -2/3)$). This explicit definition provides sufficient constraints for the model to output calibrated scalars without needing examples.
>     * **Avoiding Bias:** Few-shot prompting in regression tasks can bias the model to output only the specific values seen in the examples (e.g., clustering around 0.5 or 0.8). Zero-shot prompting with a continuous scale instruction encourages the model to utilize the full range of the interval $[-1, 1]$ based on the semantic intensity of the text.
>
> **Q8**
>
> We utilized GPT-3.5-Turbo for the primary Amazon experiments because it provided a necessary balance of computational efficiency and semantic capability for processing large-scale longitudinal corpora at the time of the study. However, the LLmFPCA-detect framework is inherently model-agnostic; the embedding module $\Phi$ is designed to be swappable, allowing the pipeline to leverage newer, more capable models without structural changes to the downstream functional analysis.
>
> To address concerns regarding model sensitivity, we conducted a comparative analysis using the Wikipedia dataset. We specifically evaluated the consistency of embeddings generated by GPT-4o-mini against those from Claude 3 Haiku using identical prompts. The results demonstrated strong cross-model alignment, yielding Pearson correlations of 0.66 for toxicity scores and 0.77 for aggression scores. This confirms that the longitudinal signals captured by our framework are robust to the specific choice of labeler and are not artifacts of a single model architecture.
>
> (continued below)

---

> ### Author Response · Authors · 2025-11-27
> **Response to Anon. Reviewer hBza (Part 5/6)**
>
> **Q9**
>
> Table 6 is present in Section A of the Appendix. It presents the distribution of dominant customer intents—such as product quality, fit, and value perceptions—across the time windows.
>
> **Q10**
>
> In our analysis of the Amazon Reviews dataset, LLmFPCA-detect identified a total of 228 unique anomalous users. As detailed in Appendix B.2, we provide specific empirical $p$-values for every detected time window in the supplementary materials to ensure transparency and obtain the window wise anomalies. We applied an identical window-based detection logic to the Wikipedia dataset across five time windows to identify toxicity anomalies, further demonstrating the method's cross-domain applicability. We intend to include the numbers in the revised version for ready reference.
>
> **Q11**
>
> We did not apply the LLmFPCA-detect framework directly to BERT embeddings because high-dimensional BERT representations pose significant challenges in sparse longitudinal settings. With only a few time-stamped observations per subject, operating with 768-dimensional vectors leads to unstable covariance estimation and inflated variance in the mFPCA step.
>
> Equally important, BERT embeddings are *generic and not interpretable*: individual dimensions do not correspond to human-meaningful concepts. In contrast, GPT-based embeddings such as emotion scores or toxicity/aggression scores are low-dimensional, domain-specific, and inherently interpretable. These properties make them well suited for LLmFPCA-detect’s functional design and support our “Signal $\to$ Flag $\to$ Explanation’’ pipeline, where axes carry semantic meaning (e.g., “Joy,’’ “Anger,’’ “Toxicity’’) that directly explains flagged anomalies.
>
> While one could reduce BERT embeddings via PCA, the resulting components would be opaque linear combinations of features and would not recover domain-relevant semantics. For this reason, we use BERT with Isolation Forest as a strong content-agnostic baseline but not as a primary choice for the embedding space.
>
> **Q12**
>
> We use the English Wikipedia request–comment stream collected via the Wikipedia API, restricted to the `user_talk` and `article_talk` namespaces and filtered to comments posted between 2010–2015; from this corpus we retain all comments authored by the 925 pseudonymized users mentioned in the paper. Each retained comment contributes one observation time for that user. The dataset comprises a total of 8,255 comments and 8,252 unique time points for these 925 users. We will include these exact descriptive statistics in Section 4.2 of the final manuscript to ensure reproducibility.
>
> **Q13**
>
> For the Wikipedia experiment, toxicity and aggression scores are produced with GPT-4o-mini using a fixed, prompt-based scoring pipeline; this is stated in Table 3, where the main method is denoted “LLmFPCA-detect (gpt-4o-mini).’’ Regarding alignment with ground truth, we computed the Pearson correlation between these GPT-derived scores and the crowdsourced human annotations available in the dataset. The analysis yields correlations of 0.63 for toxicity and 0.67 for aggression.
>
> (continued below)

---

> ### Author Response · Authors · 2025-11-27
> **Response to Anon. Reviewer hBza (Part 6/6)**
>
> ## Weaknesses
>
> **W1**
>
> The rigorous formalization, complete pseudocode, and input/output definitions for these components are fully detailed in **Appendix E.2** of the appendix. We will enhance the main text by incorporating the following high-level definitions to ensure the manuscript is self-contained:
> * **Algorithm 7 (Screening):** This step constructs the empirical Cumulative Distribution Function (CDF) of the mFPC scores within a cluster. It identifies a candidate set of potential outliers by selecting subjects that fall into the extreme tails (defined by $\alpha_1$) of this empirical distribution.
> * **Algorithm 8 (Confirmation):** This step rigorously tests the screened candidates against a held-out calibration set. It computes empirical $p$-values for the deviation magnitude and retains only those subjects whose deviation is statistically significant at the target level $\alpha$ (after multiple testing correction), thereby ensuring false-positive control.
>
> In the revision we will make sure to add a high-level formalization and an outline of our approach for better readability.
>
> **W2**
>
> We emphasize that our goal is to develop methodology for analyzing sparse longitudinal texts—an important yet largely unexplored problem, despite the routine availability of such data. While our framework builds on established tools, the novelty lies in the theoretically supported and rigorous adaptation and integration of these components into a coherent pipeline specifically designed for the challenges of Sparse Longitudinal (SL) text data. In this setting, standard methods either fail outright or collapse the trajectories into static summaries, thereby discarding the dynamic structure that is central to the scientific questions of interest. Our LLmFPCA-detect framework directly addresses these limitations. For these reasons, we respectfully disagree that our contribution is merely a combination of existing algorithms. Below, we outline how our approach differs from—and advances—the state of the art:
>
> * **Unified end-to-end framework:** LLmFPCA-detect is, to our knowledge, the first framework to unify advanced text embeddings with functional data analysis (FDA) tools—specifically mFPCA—in an end-to-end pipeline designed for SL text data. Although FDA methods are well established, they require numeric functional representations and cannot be directly applied to raw text in an interpretable manner. Furthermore, our framework handles heterogeneity in both subject trajectories and baseline static covariates and introduces a new anomaly detection procedure for heterogeneous trajectories that comes with significance guarantees—an aspect that, to the best of our knowledge, has not been addressed in the FPCA literature. Existing text-analysis pipelines do not natively support dynamic modeling over sparse longitudinal trajectories, making our approach fundamentally distinct.
> * **Explicit design for sparsity and irregularity:** Most existing models—such as topic models, intent classifiers, or sequence-based deep learning methods—assume relatively dense or regularly spaced observations. LLmFPCA-detect is explicitly built for sparsity and irregular sampling at the subject level, recovering smooth trajectories from subjects with few, irregularly spaced observations via the Karhunen-Loève expansion.
> * **Interpretability with Statistical Guarantees:** LLmFPCA-detect operates in a fully unsupervised manner while maintaining high interpretability. Unlike heuristic deep-learning detectors, we provide a theoretical foundation for our anomalies: we employ a novel two-stage screening and calibration procedure (Algorithm 2) that provides **Type-I error control** (Theorem E.1), which is new in the FDA literature. Furthermore, we incorporate dynamic keyword profiling to extract underlying intents behind anomalous windows—something not directly enabled by conventional topic models.
> * **Modularity and extensibility:** The framework is designed to be modular, allowing for substitution of emotion embeddings with other types (e.g., semantic, topic-based, or intent-based embeddings). This makes LLmFPCA-detect adaptable across domains—as demonstrated by our experiments on both Amazon Reviews (Emotions) and Wikipedia Talk Pages (Toxicity)—without sacrificing the interpretability or tractability advantages not shared by most “black-box’’ deep learning models.
>
> **W3**
>
> Please refer to our detailed responses in Questions 7 through 13 above, where we address the experimental setup, embedding choices, data selection, and model comparisons in detail.

---

### Official Review · Reviewer_ANG3 · 2025-10-31

**Soundness:** 3
**Presentation:** 2
**Contribution:** 3
**Rating:** 4
**Confidence:** 3

**Summary:**

This paper proposes LLMFPCA-DETECT, a novel framework combining LLM-based embeddings with multivariate Functional Principal Component Analysis (mFPCA) to analyze sparse longitudinal (SL) textual data—datasets where individuals generate text intermittently and irregularly over time (e.g., reviews, social media posts, medical notes).

**Strengths:**

1. The combination of LLM embeddings and functional data analysis (mFPCA) is both creative and technically elegant. It bridges modern NLP with classical statistical modeling, addressing an underexplored setting—sparse, irregular longitudinal text.
2. The paper clearly defines the challenge of SL texts, distinguishing them from standard time series or dense sequences. The motivation for type-I-controlled anomaly detection in irregular textual data is compelling.

**Weaknesses:**

1. The empirical evaluation mainly compares against Isolation Forests (on BERT and GPT embeddings). This feels weak given the paper’s ambition.
2. The performance hinges on the quality of LLM-derived embeddings (e.g., emotion or toxicity scores). However, prompt variability or model drift could affect reproducibility.
3. mFPCA requires covariance estimation and eigen-decomposition, which may scale poorly with large numbers of subjects or long trajectories.

**Questions:**

1. Can this approach generalize to dense sequences or multimodal signals (e.g., text + physiological data)?
2. Are there theoretical guarantees for cluster recovery under sparse observation settings?

---

> ### Author Response · Authors · 2025-11-26
> **Response to Anon. Reviewer ANG3 (Part 1/3)**
>
> We sincerely thank you for the detailed and thoughtful review, as well as for your questions, comments, and feedback, which we believe will substantially improve the manuscript. Below, we present our rebuttal, addressing the questions first, followed by the identified weaknesses in their original order, and we would be grateful for the opportunity to respond to any additional questions or concerns during the discussion period.
>
> (continued below)

---

> ### Author Response · Authors · 2025-11-26
> **Response to Anon. Reviewer ANG3 (Part 2/3)**
>
> ## Questions
>
> **Q1**
>
> Yes, LLmFPCA-detect is inherently designed to generalize to both dense sequences and multimodal settings, as its mathematical foundation is agnostic to the data source once mapped to a numeric functional space.
>
> * **Dense Sequences:** While our paper focuses on the more challenging *sparse* case (where subjects have few, irregular observations), the functional data analysis (FDA) framework applies equally well to dense, regularly sampled grids. In dense settings, the estimation of covariance surfaces and principal component scores becomes simpler and more accurate, as the algorithm no longer needs to borrow information across subjects to the same extent required for sparse data.
> * **Multimodal Signals:** The framework explicitly models multivariate trajectories $\boldsymbol{Y}_i(t) \in \mathbb{R}^p$. This $p$-dimensional vector can naturally represent a hybrid signal where some dimensions are LLM-derived text embeddings (e.g., emotion scores) and other directions measure something else (e.g., heart rate or blood pressure). The use of **multivariate FPCA** (mFPCA) is specifically advantageous here, as it jointly captures the variations in possibly correlated modalities— for instance, textual “Anxiety" and a rise in physiological signals, which independent univariate models would miss. We appreciate your observation and will make sure to add this discussion in our revision.
>
> **Q2**
>
> While we support our proposed methodology with theory, the primary goal of this paper is to design the LLmFPCA-detect pipeline—a versatile framework applicable to any dataset consisting of sparse longitudinal texts and intended as a practical tool for learning representations in such complex data structures, in line with the ICLR theme. To keep the theoretical analysis simplified and tractable, our results assume fully observed trajectories which helps us motivate our approach without having to deal with excessive notations. As we explicitly note in the Conclusion section, extending these guarantees to the setting where mFPCA estimates are derived directly from sparse observations is an important direction for future work.
>
> ## Weaknesses
>
> **W1**
>
> Since our paper focuses on complex sparse longitudinal text data, there is a lack of existing methods in the literature that directly address this problem, and thus few suitable methods in the literature for comparison. Given this, we have included substantial empirical validation through experiments spanning multiple domains, comparisons against the strongest available baselines, and rigorous sensitivity simulations.
>
> * **Cross-Domain Validation:** We demonstrate generalization across multiple real world datasets. We apply our method on the Amazon Reviews data and the Wikipedia Talk Page corpus (Section 4.2). This validates the method on completely different signal types (toxicity/aggression vs. emotion) and for the Wikipedia dataset, allows for ground-truth verification using human annotations as well.
> * **Baseline Comparisons:** For the Wikipedia dataset, where we have surrograte ground truth, we explicitly compared LLmFPCA-detect against state-of-the-art unsupervised baselines, specifically **Isolation Forest** using **BERT embeddings** and **GPT-derived scores** (Table 3). Our method achieved superior F1 scores (e.g., 0.58 vs 0.41 in TW1), demonstrating that modeling with the *functional principal component scores* provides detection power that point-wise methods miss.
> * **Rigorous Simulation Study:** As detailed in **Appendix B.2**, we conducted extensive simulations ($S=50$ replicates) to stress-test the algorithm under varying levels of sparsity (truncated Poisson sampling). We quantified the stability of anomaly detection across clusters, identifying that while recall is robust ($>0.8$) for distinct behaviors, subtler anomalies (Cluster 2) require the specific window-based testing we developed.
> * **Downstream Predictive Utility:** Beyond anomaly detection, we validated the utility of the learned representations in a supervised forecasting task (Section 4.1). The mFPC scores improved the prediction of sudden customer rating drops by **12.4%** in accuracy and **20.8%** in ROC-AUC compared to standard aggregated historic ratings which ignores the dynamic aspect.
>
> (continued below)

---

> ### Author Response · Authors · 2025-11-26
> **Response to Anon. Reviewer ANG3 (Part 3/3)**
>
> **W2.**
>
> We are aware of this concern, and have therefore taken care to ensure that, for any given LLM, we condition our analysis on its outputs. Once the LLM and prompt are fixed, the outputs are deterministic and introduce no additional variability. In addition, we have stress-tested the robustness of our approach across different LLMs, and in the revision we will include comparisons under alternative prompts. As language modeling continues to evolve rapidly, our goal is to leverage these developments to build methodology for analyzing sparse longitudinal texts—an important problem that has received little attention to date. While we acknowledge that the quality of the analysis may be sensitive to prompt choice and the specific LLM used, the innovation of our work lies in demonstrating that advances in LLMs can be systematically incorporated to solve this statistical problem. We describe these considerations below and emphasize that, moving forward, one can readily substitute any state-of-the-art LLM within the LLmFPCA-detect pipeline.
>
> * **Conditioning on LLM outputs:** Our framework explicitly conditions om the LLM outputs, therefore treating the map $\Phi$ as fixed. We enforce this by fixing the temperature to 0 (greedy decoding) and using fixed seeds. This ensures that for a given model version, the mapping from text to numeric vector is stable and reproducible.
> * **Cross-Model Sensitivity Analysis:** We evaluate the robustness of our embeddings against model variation by conducting experiments on the Wikipedia comments dataset. We compared toxicity and aggression scores generated by two distinct models, **Claude 3 Haiku** and **GPT-4o-mini**, using identical prompts. The results showed strong alignment, with Pearson correlations of **0.66** for toxicity scores and **0.77** for aggression scores. This indicates that the extracted signals are reasonably consistent across different LLM architectures.
> * **Modularity:** LLmFPCA-detect is designed as a modular framework. While we demonstrate performance using specific models (GPT-3.5/4o), the pipeline is agnostic to the embedder. As open-source models improve, they can be swapped into the embedding module $\Phi$ without altering the downstream functional analysis or anomaly detection logic.
>
> **W3**
>
> Theoretically and empirically, the computational complexity of our framework matches the standard complexity of estimating mean functions and eigenfunctions in the FDA literature, making the method scalable to large numbers of subjects and/or time points. As detailed in Algorithm 1, the multivariate FPCA step operates on the covariance matrices obtained from the *univariate* FPCA components and leverages linear algebra to efficiently translate univariate results into multivariate eigenfunctions. Consequently, the overall computational burden is essentially governed by the complexity of PACE—the established algorithm for univariate FPCA—ensuring that our framework remains practical even in large-scale settings.
>
> For reference in the Wiki data we use the English Wikipedia request–comment stream collected via the Wikipedia API, restricted to the `user_talk` and `article_talk` namespaces and filtered to comments posted between 2010–2015; from this corpus we retain all comments authored by the 925 pseudonymized users mentioned in the paper. Each retained comment contributes one observation time for that user. The dataset comprises a total of 8,255 comments and 8,252 unique time points for these 925 users, and the FPCA step step takes only a few minutes to run for the entire population with decreasing run times for the clusters.

---

### Official Review · Reviewer_4qm1 · 2025-11-01

**Soundness:** 3
**Presentation:** 2
**Contribution:** 2
**Rating:** 4
**Confidence:** 2

**Summary:**

The paper addresses the challenge of analyzing sparse longitudinal (SL) textual data—settings where individuals generate text at irregular times (e.g., customer reviews, social media posts, or clinical notes). Existing text mining methods (e.g., BERT-based embeddings or time-series models) assume dense, regular sampling and thus fail to handle the irregular timing, heterogeneity, and noise in SL data. The proposed LLmFPCA-detect framework integrates Large Language Models (LLMs) with multivariate functional principal component analysis (mFPCA) to enable clustering and anomaly detection on SL text streams.

**Strengths:**

The paper presents a novel integration of LLM-derived embeddings with statistical functional analysis via mFPCA. It introduces outlier detection, featuring a two-stage anomaly calibration procedure (Algorithms 2 and 3) that employs sample splitting and multiple-comparison control. The authors also provide empirical studies demonstrating the effectiveness of the proposed approach.

**Weaknesses:**

The novelty of the work is not very clear, as the proposed method appears to combine elements of existing algorithms rather than introducing a fundamentally new approach.

Moreover, the paper lacks strong theoretical justification. There are some guarantees in appendix, but their scopes seem limited.

Given the absence of strong theoretical results, the empirical validation also feels limited. A broader set of simulations or comparisons would strengthen the paper’s overall contribution.

**Questions:**

Could the authors more clearly articulate the novelty of the proposed method? In its current form, the approach appears to be a combination of existing algorithms, and the unique conceptual or technical contribution is not immediately evident.

The overall algorithmic pipeline is quite long, and several key components are deferred to the appendix. It would be helpful to include a concise “prototype” version of the algorithm in the main text—summarizing the core steps and key ideas—so that readers can quickly grasp the main workflow without referring to supplementary materials.

---

> ### Author Response · Authors · 2025-11-26
> **Response to Anon. Reviewer 4qm1 (Part 1/3)**
>
> We sincerely thank you for the detailed and thoughtful review, as well as for your questions, comments, and feedback, which we believe will substantially improve the manuscript. Below, we present our rebuttal, addressing the questions first, followed by the identified weaknesses in their original order, and we would be grateful for the opportunity to respond to any additional questions or concerns during the discussion period.
>
> (continued below)

---

> ### Author Response · Authors · 2025-11-26
> **Response to Anon. Reviewer 4qm1 (Part 2/3)**
>
> ## Questions
>
> **Q1**
>
> We emphasize that our goal is to develop methodology for analyzing sparse longitudinal texts—an important yet largely unexplored problem, despite the routine availability of such data. While our framework builds on established tools, the novelty lies in the theoretically supported and rigorous adaptation and integration of these components into a coherent pipeline specifically designed for the challenges of Sparse Longitudinal (SL) text data. In this setting, standard methods either fail outright or collapse the trajectories into static summaries, thereby discarding the dynamic structure that is central to the scientific questions of interest. Our LLmFPCA-detect framework directly addresses these limitations. For these reasons, we respectfully disagree that our contribution is merely a combination of existing algorithms. Below, we outline how our approach differs from—and advances—the state of the art:
>
> * **Unified end-to-end framework:** LLmFPCA-detect is, to our knowledge, the first framework to unify advanced text embeddings with functional data analysis (FDA) tools—specifically mFPCA—in an end-to-end pipeline designed for SL text data. Although FDA methods are well established, they require numeric functional representations and cannot be directly applied to raw text in an interpretable manner. Furthermore, our framework handles heterogeneity in both subject trajectories and baseline static covariates and introduces a new anomaly detection procedure for heterogeneous trajectories that comes with significance guarantees—an aspect that, to the best of our knowledge, has not been addressed in the FPCA literature. Existing text-analysis pipelines do not natively support dynamic modeling over sparse longitudinal trajectories, making our approach fundamentally distinct.
>
> * **Explicit design for sparsity and irregularity:** Most existing models—such as topic models, intent classifiers, or sequence-based deep learning methods—assume relatively dense or regularly spaced observations. LLmFPCA-detect is explicitly built for sparsity and irregular sampling at the subject level, recovering smooth trajectories from subjects with few, irregularly spaced observations via the Karhunen-Loève expansion.
>
> * **Interpretability with Statistical Guarantees:** LLmFPCA-detect operates in a fully unsupervised manner while maintaining high interpretability. Unlike heuristic deep-learning detectors, we provide a theoretical foundation for our anomalies: we employ a novel two-stage screening and calibration procedure (Algorithm 2) that provides **Type-I error control** (Theorem E.1), which is new in the FDA literature. Furthermore, we incorporate dynamic keyword profiling to extract underlying intents behind anomalous windows—something not directly enabled by conventional topic models.
>
> * **Modularity and extensibility:** The framework is designed to be modular, allowing for substitution of emotion embeddings with other types (e.g., semantic, topic-based, or intent-based embeddings). This makes LLmFPCA-detect adaptable across domains—as demonstrated by our experiments on both Amazon Reviews (Emotions) and Wikipedia Talk Pages (Toxicity)—without sacrificing the interpretability or tractability advantages not shared by most “black-box’’ deep learning models.
>
> In summary, LLmFPCA-detect offers a novel and practical solution to a currently unmet need: analyzing unsupervised, sparse, and irregular longitudinal text data in a way that is interpretable, generalizable, and motivated with theory.
>
> (continued below)

---

> ### Author Response · Authors · 2025-11-26
> **Response to Anon. Reviewer 4qm1 (Part 3/3)**
>
> **Q2**
>
> We appreciate this suggestion to improve readability. In the final manuscript, we will introduce a high-level summary and a pictorial representation of the “Master Pipeline’’ in the main text. This will allow readers to visualize the complete architecture at a glance, summarizing the core workflow as follows:
>
> * **Step 1: Embedding.** Map raw texts to fixed numeric vectors via LLM prompting.
> * **Step 2: Global Representation.** Estimate global mFPC scores using the pooled population.
> * **Step 3: Segmentation.** Cluster subjects into groups using the mFPC scores and baseline covariates jointly.
> * **Step 4: Local Refinement.** Re-fit mFPCA parameters separately within each cluster for cluster-specific trajectory reconstructions.
> * **Step 5: Subject Anomaly Detection.** Identify globally anomalous subjects via tail screening and calibrated testing.
> * **Step 6: Window Localization.** Pinpoint specific anomalous time windows for flagged subjects.
> * **Step 7: Interpretation.** Generate dynamic keyword profiles for confirmed anomalous windows using LLMs.
>
> ## Weaknesses
>
> **W1**
>
> See Q8.
>
> **W2**
>
> Thank you for raising this point, which allows us to emphasize the central novelty of our work: an end-to-end pipeline for analyzing sparse longitudinal text data. Such data are common across applications, yet are often discarded because of the lack of tools for extracting meaningful representations from unstructured text under sparsity. By combining the representational power of LLMs with the flexibility of functional data analysis for sparse trajectories, our framework provides a fully unsupervised pipeline for representation learning and anomaly detection in this setting.
>
> While our approach is supported by theory, developing new theoretical machinery is not the primary goal of this paper. We intentionally keep the theoretical component simple and tractable, focusing on motivating and validating the *novel unsupervised learning* elements of our framework, including cluster recovery in the presence of anomalies and principled anomaly detection with statistical significance under sparse longitudinal designs. We do not re-derive convergence rates for estimating smooth trajectories from sparse data, as these results are well established in the Functional Data Analysis literature. More intricate theoretical developments for this framework represent an important direction for future work.
> We acknowledge this in the Conclusion that bridging these two theoretical modules—deriving end-to-end bounds that explicitly propagate the sparse estimation error into the clustering bounds—is a non-trivial extension for future work. However, our simulation studies (Appendix B.2) empirically validate that the method remains robust under realistic sparsity levels.
>
> **W3**
>
> Since our paper focuses on complex sparse longitudinal text data, there is a lack of existing methods in the literature that directly address this problem, and thus few suitable baselines for comparison. Given this, we have included substantial empirical validation through experiments spanning multiple domains, comparisons against the strongest available baselines, and rigorous sensitivity simulations.
>
> * **Cross-Domain Validation:** We demonstrate generalization across multiple real world datasets. We apply our method on the Amazon Reviews data and the Wikipedia Talk Page corpus (Section 4.2). This validates the method on completely different signal types (toxicity/aggression vs. emotion) and for the Wikipedia dataset, allows for ground-truth verification using human annotations as well.
> * **Baseline Comparisons:** For the Wikipedia dataset, where we have surrogate ground truth, we explicitly compared LLmFPCA-detect against state-of-the-art unsupervised baselines, specifically **Isolation Forest** using **BERT embeddings** and **GPT-derived scores** (Table 3). Our method achieved superior F1 scores (e.g., 0.58 vs 0.41 in Time Window 1 (TW1)), demonstrating that modeling with the *functional principal component scores* provides detection power that point-wise methods miss.
> * **Rigorous Simulation Study:** As detailed in **Appendix B.2**, we conducted extensive simulations ($S=50$ replicates) to stress-test the algorithm under varying levels of sparsity (truncated Poisson sampling). We quantified the stability of anomaly detection across clusters, identifying that while recall is robust ($>0.8$) for distinct behaviors, subtler anomalies (Cluster 2) require the specific window-based testing we developed.
> * **Downstream Predictive Utility:** Beyond anomaly detection, we validated the utility of the learned representations in a supervised forecasting task (Section 4.1). The mFPC scores improved the prediction of sudden customer rating drops by **12.4%** in accuracy and **20.8%** in ROC-AUC compared to standard aggregated historic ratings which ignores the dynamic aspect.

---

### Official Review · Reviewer_R6T7 · 2025-11-01

**Soundness:** 3
**Presentation:** 2
**Contribution:** 2
**Rating:** 6
**Confidence:** 2

**Summary:**

This paper presents a pipeline for the analysis of anomalies in sparse longitudinal texual data. The raw texts are converted to numerical vectors with prompting an LLM, and the mFPCA is used to extract the mFPC scores, which are the basis for later postprocessing steps, including clustering, anomaly detection, and keyword extraction. The proposed pipeline is demonstrated on an amazon review dataset and a Wikipedia request-comment dataset. Interesting patterns are shown by the clusters and anomalies are interpreted with the help of the extracted keywords. It is obvious that the proposed pipeline could be appiled to many more scenarios.

**Strengths:**

* The pipeline decribed in the paper shows an end2end procedure to analyze textual corpus with timestamps. The highlight of the algorithm is the interpretability of the anomalies found, which is very useful in real scenarios.
* The application of the pipeline is demostrated on two use cases in detail. The many qualitative results are interesting to read.
* The description of the algorithms in the many Alg blocks are clear and helpful for readers to understand.

**Weaknesses:**

* It is a complex system and have many steps. The design of some important steps are only presented but not justified. So, it reads somewhat like product white paper. (1) Why mFPCA instead of uFPCA, and what is the benefit? (2) Is Alg. 2 a standard way of anomaly detection (if so, what's the citation), or a novel one (if so, why is it a good design)? (3) Why clustering is required at all, and how to determine the number of clusters?
* The evaluation are mostly qualitative, and very few baselines are compared with.

**Questions:**

* At L157, the modeling uses "cluster-specific mean function" and "shared eigenfunctions across clusters", while at L240, cluster-specific mean and eigenfunctions are used.
* L191. Why is the output of LLM deterministic, and why does this determinism matter? Generally, LLM outputs are sampled with randomness.
* Why is the *functional* version of PCA required? What if the time axis is dropped (or divide into groups by time window if localization is wanted), and do regular PCAs; Will anomalies still identifiable?
* What does the boldface mean in Tab. 1?
* L317. Could you elaborate on how the baseline "by mean Automobile rating" works? Beside this baseline, are there more baselines?
* The lower two rows of Tab. 3 is identical. Is this a typo or by chance?
* L413. Which method is referred as the "state-of-the-art", any citation for the method?

---

> ### Author Response · Authors · 2025-11-26
> **Response to Anon. Reviewer R6T7 (Part 1/4)**
>
> We sincerely thank you for the detailed and thoughtful review, as well as for your questions, comments, and feedback, which we believe will substantially improve the manuscript. Below, we present our rebuttal, addressing the questions first, followed by the identified weaknesses in their original order, and we would be grateful for the opportunity to respond to any additional questions or concerns during the discussion period.
>
> (continued below)

---

> ### Author Response · Authors · 2025-11-26
> **Response to Anon. Reviewer R6T7 (Part 2/4)**
>
> ## Questions
>
> **Q1:**
>
> Thank you for this careful observation, which allows us to clarify that we indeed adopt a two-stage “coarse-to-fine’’ design for the implementation of our pipeline in order to do justice to the potential heterogeneity in the dataset. The assumption of shared eigenfunctions is primarily technical, supporting the theory behind our approach rather than serving as a practical requirement. Even our theoretical framework can, in principle, accommodate heterogeneous eigenfunctions across clusters, but we avoid this extension solely for the sake of notational and analytical simplicity. While this assumption is introduced mainly to simplify the theoretical analysis and is not a primary goal of the paper, our methodological framework is fully capable of accommodating, and in fact benefits from, cluster-specific eigenfunctions in the refinement stage. Shared eigenfunctions act as a starting point, since clustering is an unsupervised problem and, in the absence of knowledge of cluster labels, it is not possible to estimate cluster-specific eigenfunctions to begin with. Later, once the clusters have been estimated, the estimation of the eigenfunctions can be further refined for each cluster. In practice, even if the eigenfunctions are indeed shared, estimating them within each cluster does not contradict this assumption. However, estimating them in a cluster-specific manner offers advantages, as detailed in the outline of our two-stage approach below.
>
> 1. **Stage 1: Global Segmentation (Shared Eigenfunctions).** In the absence of known cluster labels, we apply FPCA to the entire cohort to obtain a common latent score space in which all heterogeneous subjects can be projected. These FPCA scores are then augmented with other covariates in the clustering subroutine in Algorithm 5 to obtain the initial cluster assignments. While this step assumes shared eigenfunctions (Sec 2, L157), this assumption is made primarily for simplification of the theory and can be extended to accommodate cluster-specific eigenfunctions.
>
> 2. **Stage 2: Local Refinement (Cluster-Specific Eigenfunctions).** After the clusters $\hat{\mathcal{C}}_k$ have been estimated, we refit FPCA *separately* within each cluster to capture group-specific dynamics as detailed in Algorithm 6. This yields cluster-specific eigenfunctions $\hat{\psi}^k_m$ and means $\hat{\mu}_k$, which substantially reduce reconstruction error for the “clean’’ cohort and, in turn, markedly enhance the sensitivity and reliability of the downstream anomaly detection procedures (Algorithms 2 & 3).
>
> We will revise Sec. 3 to explicitly highlight this transition from the global model (used for segmentation) to the refined local models (used for detection).
>
> **Q2:**
>
> You are absolutely right that, under typical settings, LLM outputs are stochastic due to probabilistic token sampling strategies such as nucleus sampling or the use of non-zero temperature values. By the deterministic assumption, we mean that our analysis is conducted conditional on the LLM outputs, since our inferential goals concern the randomness among subjects rather than variability in how texts are scored. Because the randomness in LLM generation is independent of the data, conditioning on the outputs is natural and avoids unnecessary noise. To achieve this, we set the temperature to 0 (greedy decoding) and fix the random seed, making the LLM outputs reproducible. This ensures that, once the LLM outputs are obtained, all remaining randomness in the pipeline comes solely from subject sampling and their observation times. To summarize, this determinism arising from conditioning on the LLM outputs matters for two reasons:
>
> 1. **Invariance to randomness in text scoring (Ensuring validity of the mapping $\Phi$):** Our framework defines the embedding step as a mathematical function $\Phi: \mathcal{K} \to \mathbb{R}^p$, where a specific text input $K$ must map to a unique numeric vector $Y$. If the LLM output were stochastic, $\Phi(K)$ conditional on $K$, would be random rather than deterministic. This would introduce *artificial randomness*—generated by the measurement tool (the LLM)—rather than the subject. By enforcing determinism through conditioning on the LLM outputs, we ensure that any variation in the trajectory $Y_i(t)$ arises solely from the variability among **subject's evolving behavior** or measurement errors $\eta_i(t)$, not from the randomness of our feature extractor.
>
> 2. **Reproducibility:** For the scientific validity of our anomaly detection benchmarks (e.g., Tab 1 and Tab 3), it is essential that other researchers can replicate our exact numeric trajectories given the same raw text corpus. Stochastic embeddings would make the cluster assignments and anomaly flags unstable across different runs.
>
> We clarify in Sec 3 that we explicitly use greedy decoding (temp = 0) so that the LLM operates as a deterministic scoring function rather than a stochastic generative agent.
>
> (continued below)

---

> ### Author Response · Authors · 2025-11-26
> **Response to Anon. Reviewer R6T7 (Part 3/4)**
>
> **Q3:**
>
> We use functional (mFPCA) rather than ordinary PCA because subjects are observed at *sparse, irregular, subject-specific* time points, often with only a handful of observations per trajectory. In such sparse longitudinal designs, there is no common time grid on which all subjects can be stacked into vectors without introducing strong, ad hoc pre-processing. Moreover, the number of time points can change from subject to subject, and the measurements per subject are ordered in time and therefore cannot be stacked as regular multivariate objects. Functional PCA is specifically designed for this setting: it treats $\{Y_i(T_{ij})\}$ as noisy evaluations of an underlying smooth trajectory $X_i(t)$ and recovers the mean and covariance in continuous time, then obtains low-dimensional scores via the Karhunen–Loève expansion (Yao et al., 2005; Happ & Greven, 2018).
>
> If we drop time altogether, there is no straightforward way to obtain multivariate representations per subject, and we also lose the dynamic trends in each subject’s behavior, which is a primary goal of this paper. If instead we bin time into smaller windows and apply standard PCA to some vector representation per window, we face issues such as:
>
> 1. **Irregular design and missingness.** Different subjects contribute to different time windows, and under sparse designs, many subjects may be missing in several windows.
>
> 2. **Loss of temporal resolution.** Our anomalies are defined as deviations $a_i(t)\neq 0$ over subsets of time $T_0$; these deviations enter linearly into the mFPC scores and shift their expectations in a way we exploit for detection. Aggregating within wide windows dilutes short or localized anomalies and makes them harder to distinguish from regular variability.
>
> In principle, very strong anomalies that affect all time windows could still be detectable with a windowed non-functional PCA. However, for sparse longitudinal trajectories, functional PCA provides (i) a principled way to handle irregular sampling without pre-smoothing of trajectories, (ii) smooth, interpretable eigenfunctions that capture time-evolving modes of variation, and (iii) mFPC scores whose distribution under the null is well-characterized and directly used in our anomaly calibration theory. In the revision, we will clarify our motivation for adopting sparse mFPCA rather than a time-windowed PCA approach.
>
> **Q4:**
>
> The boldface highlights the dominant or most salient emotion dimension that characterizes the specific "pain point" or anomaly discussed in the main text. Specifically:
> * In the first example (5-star review), the bolded **-1** in the *Surprise--Anticipation* column highlights the customer's extreme surprise regarding the price ("$17.50 for a single quart is STUPID"), which contradicts the positive rating.
> * In the third example (1-star review), the bolded **-0.75** in the *Trust--Disgust* column highlights the strong signal of disgust regarding product quality ("Absolute junk!"), which drives the negative anomaly.
>
> We will update the caption of Table 1 to explicitly state: "Boldface values indicate the emotion dimension with the highest intensity or the primary driver of the anomaly discussed in the text."
>
> **Q5:**
>
> In Section 4.1, the “mean Automobile rating’’ baseline corresponds to Model A: for each user, we compute the cumulative average star rating in the Automobile category over their entire history and use this single scalar summary—together with the same structured covariates (e.g., cross-category purchase mix)—as input to a random forest classifier for predicting sudden rating drops. Model B keeps both the learning algorithm and covariates same as Model A but replaces the scalar rating summary with the mFPC emotion scores. This comparison isolates the added value of incorporating dynamic emotion trajectories for predicting rating drops, relative to historical rating aggregates that ignore temporal dynamics.
> In addition to this supervised baseline on Amazon, our anomaly-detection experiments on the Wikipedia data compare LLmFPCA-detect against Isolation Forest applied to (i) BERT embeddings and (ii) GPT-derived toxicity/aggression scores, providing further baselines in a setting with surrogate ground truth.
>
> **Q6:**
>
> This is a verified empirical finding, not a typo.
>
> **Q7:**
>
> The term “state-of-the-art’’ in this section refers to the use of Isolation Forest, a state-of-the-art anomaly detection method. While widely recognized as a standard baseline for anomaly detection, the relevant references include [1] (published as a conference paper at ICLR 2020).
>
> [1] Ruff, Lukas, et al. “Deep Semi-Supervised Anomaly Detection.” arXiv, 2020, arXiv:1906.02694.
>
> (continued below)

---

> ### Author Response · Authors · 2025-11-26
> **Response to Anon. Reviewer R6T7 (Part 4/4)**
>
> ## Weaknesses
>
> **W1:**
>
> **1. Justification for mFPCA over uFPCA.** We employ Multivariate FPCA (mFPCA) rather than Univariate FPCA (uFPCA) to capture the *joint* dependency structure between the embedding dimensions. While uFPCA treats each dimension (e.g., Joy–Sadness vs. Trust–Disgust) independently, mFPCA explicitly accounts for the covariance between them (as detailed in Appendix C.2) and gives more parsimonious representations. This is critical because emotional states or toxicity markers are rarely independent; for instance, a rise in “Anger’’ is often correlated with a drop in “Trust.’’ Capturing these multivariate objects jointly allows the model to detect anomalies that are defined by *unusual combinations* of features, which uFPCA would miss.
>
> **2. Novelty and Design of Algorithm 2.** To the best of our knowledge, Algorithm 2 is a novel contribution to the FDA literature: prior anomaly-detection methods for sparse longitudinal data do not provide significance guarantees. Detecting functional anomalies is inherently challenging, as deviations may only become apparent when the *dynamic* trajectory is considered rather than at any single time point. Our two-stage *Screening and Calibration* procedure—screening via tail probabilities and confirmation via calibration sets—addresses this challenge in a principled way and is explicitly designed to ensure Type-I error control. As established in Theorem E.1, the screened set contains the true anomalous subjects, and candidate anomalies are calibrated against clean independent data The pre-screening mechanism is flexible and can be combined with other anomaly-detection subroutines if desired.
>
> **3. Necessity of Clustering and Selection of $K$.** Clustering is a fundamental requirement to address the *heterogeneity* inherent in subject patterns (Section 2). Subjects naturally stratify into different baselines (e.g., consistently critical vs. consistently positive reviewers). Without clustering, all subjects are treated homogeneously, for example, a user in the Amazon dataset with a consistently negative sentiment might be flagged as a global anomaly simply for being pessimistic, rather than for deviating from their own behavioral norm. By defining anomalies relative to cluster-specific means (Algorithm 6), we isolate genuine deviations from diverse baseline behaviors. We determine the optimal number of clusters $K$ using the average silhouette width, as detailed in the Cluster Stability analysis in Appendix D.3.
>
> **W2:**
>
> Thank you for bringing up this point. Unsupervised learning inherently lacks ground-truth labels or outcomes, making quantitative evaluation challenging. Because our focus is on unsupervised learning and inference, our scope for quantitative evaluation is limited and some components of the exposition (e.g., keyword traces and case studies) are necessarily qualitative. That said, we have taken care to ensure that the overall evaluation is not primarily qualitative and includes quantitative assessments in the simulations and wherever feasible in the real applications including:
>
> * On Amazon, we conduct a supervised forecasting experiment where replacing the mean historical Automobile rating with past emotion mFPC scores in an otherwise identical random forest model improves accuracy, F1 score, and ROC–AUC, quantifying the added value of trajectory-based features over aggregated information in time.
> * On Wikipedia, we use the human-annotated toxicity and aggression scores as surrogate ground truth. We then generate pseudo–ground-truth anomaly labels by applying Isolation Forest to these human scores together with structured covariates. Using these labels, we evaluate LLmFPCA-detect by reporting F1 scores across five time windows and compare it against Isolation Forest applied to GPT-derived scores and to BERT embeddings. Across all windows, LLmFPCA-detect consistently achieves higher F1 scores.
>
> In simulations, we also report quantitative cluster stability via bootstrapped Jaccard indices for both domains and anomaly stability under realistic sparsity via subsampling experiments and formal tests on recall and detection probabilities. We will clarify this mix of quantitative metrics and qualitative illustrations in the revision so that the evaluation strategy is more explicit.

---

### Author Response · Authors · 2025-12-04
**Author Final Remarks**

Our paper introduces LLmFPCA-detect, an unsupervised anomaly detection method for heterogeneous, sparsely observed longitudinal text data common in online platforms, where no dedicated tools exist. Combining the representational power of large language models with multivariate sparse functional principal component analysis, LLmFPCA-detect’s flexible pipeline can adapt seamlessly to diverse domains, from emotion trajectories to toxicity monitoring. Its off-the-shelf design and open-ended scope make it both a practical, widely applicable solution and a pioneering approach with theoretical grounding. We thank the reviewers and summarize the main points addressed in our rebuttal below.

* **Novelty and unified end-to-end framework:** We clarified that our work is not merely a combination of existing algorithms but a novel, unified pipeline designed specifically for the challenges of sparse, irregular text streams where standard methods fail. LLmFPCA-detect is the first framework to unify advanced text embeddings with functional data analysis specifically for sparse longitudinal data, enabling interpretable analysis without collapsing trajectories into static summaries: a capability not enabled by conventional topic models or black-box detectors.

* **Theoretical guarantees and statistical rigor:** Addressing concerns about theoretical depth, we highlighted that our method provides more than just heuristic detection. We employ a novel two-stage screening and calibration procedure (Algorithm 2) that provides formal Type-I error control (Theorem E.1), a contribution that is new to the functional data analysis (FDA) literature and ensures statistical reliability in unsupervised settings.

* **Baselines and quantitative evaluation:** We strengthened our empirical evaluation by comparing against strong baselines in settings with surrogate ground truth. On the Wikipedia dataset, LLmFPCA-detect consistently achieved higher F1 scores than Isolation Forest applied to both BERT embeddings and GPT-derived scores, demonstrating that modeling the functional trajectory provides detection power that point-wise methods miss.

* **Handling heterogeneity and robustness:** We clarified our "coarse-to-fine" design, where shared eigenfunctions are used only for initialization, while the final local refinement stage explicitly models cluster-specific dynamics. Furthermore, we demonstrated robustness to the choice of LLM, showing that scores from Claude 3 Haiku and GPT-4o-mini were strongly aligned, and ensured reproducibility by conditioning on deterministic LLM outputs.

While there was no active discussion phase, **we believe the rebuttal provided comprehensive responses to all reviewer questions and fully addressed the raised concerns**, and we trust the final decision will reflect the paper's methodological novelty and empirical merits.

Sincerely,

Authors

---

### Meta-Review · Area_Chair_q7fq · 2026-01-02

**Summary:**

The paper addresses anomaly detection in sparse, irregular longitudinal text and proposes a multi-step pipeline that combines LLM-based scoring with sparse multivariate FPCA, clustering, and statistical screening.

Reviewers agree that the paper tackles a timely problem and presents a technically coherent and carefully implemented framework. The main concern is limited conceptual novelty: the contribution is primarily an integration of existing components rather than a clearly articulated new algorithmic or statistical insight. Although the authors highlight a screening/calibration procedure with theoretical guarantees, its scope, assumptions, and general significance are not sufficiently clear or convincing as a standalone contribution. Furthermore, most reviewers raised concerns regarding experiments. The evaluation relies heavily on proxy labels and a narrow set of baselines, provides limited quantitative validation for key claims, and does not fully establish robustness or the necessity of the whole pipeline, leaving uncertainty about the generality and strength of the empirical evidence.

Overall, while the work is promising and well-motivated, stronger empirical evaluation and ablations, as well as a sharper articulation of the core contribution, would be needed to support a recommendation for acceptance.

**Reviewer Concerns:**

R6T7: Limited justification for a complex, multi-stage approach. Partially addressed.

R6T7, 4qm1, ANG3: Evaluation strength and baseline coverage are limited. Partially addressed.

4qm1, hBza: Unclear technical novelty beyond combining existing components. Partially addressed.

4qm1: Insufficient theoretical justification. Not convincingly addressed.

ANG3: Reliability of LLM-derived features. Partially addressed.

ANG3: Scalability and computational feasibility. Not convincingly addressed.

R6T7, hBza: Clarity and completeness of presentation. Addressed

**Reviewer Scores:**

R6T7: Likely would have stayed at 6.

4qm1: Likely would have stayed at 4.

ANG3: Likely would have stayed at 4.

hBza: Likely would have stayed at 4.

---

### Decision · Program_Chairs · 2026-01-26

Reject